# Sensitivity studies with the Regional Climate Model COSMO-CLM 5.0 over the CORDEX Central Asia Domain

Emmanuele Russo[1,2,3], Ingo Kirchner[1], Stephan Pfahl[1], Martijn Schaap[4,1], and Ulrich Cubasch[1]

[1]Institute for Meteorology, Freie Universität Berlin, Carl-Heinrich-Becker-Weg 6-10, 12165, Berlin, Germany
[2]Climate and Environmental Physics, Physics Institute, University of Bern, Sidlerstrasse 5, 3012, Bern, Switzerland
[3]Oeschger Centre for Climate Change Research, University of Bern, Hochschulstrasse 4, 3012, Bern, Switzerland
[4]TNO Built Environment and Geosciences, Department of Air Quality and Climate, Princetonlaan 6, 3584, CB, Utrecht, The Netherlands

**Correspondence:** emmanuele.russo@met.fu-berlin.de

**Abstract.** Due to its extension, geography and the presence of several under-developed or developing economies, the Central Asia domain of the Coordinated Regional climate Downscaling Experiment (CORDEX) is one of the most vulnerable regions on Earth to the effects of climate changes. Reliable information on potential future changes with high spatial resolution acquire significant importance for the development of effective adaptation and mitigation strategies for the region. In this context, Regional Climate Models (RCMs) play a fundamental role.

In this paper, the results of a set of sensitivity experiments with the regional climate model COSMO-CLM version 5.0, for the Central Asia CORDEX domain, are presented. Starting from a reference model setup, general model performances is evaluated for present-days, testing the effects of singular changes in the model physical configuration and their mutual interaction on the simulation of monthly and seasonal values of three variables that are important for impact studies: near surface temperature, precipitation and diurnal temperature range. The final goal of this study is two-fold: having a general overview of model performance and its uncertainties for the considered region and determining at the same time an optimal model configuration.

Results show that the model presents remarkable deficiencies over different areas of the domain. The combined change of the albedo taking into consideration the ratio of forest fractions and the soil conductivity taking into account the ratio of liquid water and ice in the soil, allows to achieve the best improvements in model performances in terms of climatological means. Importantly, the model seems to be particularly sensitive to those parameterizations that deal with soil and surface features, and that could positively affect the repartition of incoming radiation. The analyses also show that improvements in model performances are not achievable for all domain sub-regions and variables, and they are the result of some compensation effect in the different cases. The proposed better performing configuration in terms of mean climate, leads to similar positive improvements when considering different observational datasets and boundary data employed to force the simulations. On the other hand, due to the large uncertainties in the variability estimates from observations, the use of different boundaries and the model internal variability, it has not been possible to rank the different simulations according to their representation of the monthly variability.

This work is the first ever sensitivity study of an RCM for the CORDEX Central Asia domain and its results are of fundamental importance for further model development and for future climate projections over the area.

## 1 Introduction

Regional Climate Models (RCMs) are a fundamental tool for the study of climate change, allowing to reproduce the climate system with a high quality of details and to provide information at a regional scale. Their use for future climate projections, constitutes indeed a vital resource for policy makers in their decision making under the threat of future global warming (Kim et al., 2014).

The Coordinated Regional climate Downscaling Experiment (CORDEX) (Giorgi et al., 2009) is an initiative sponsored by the World Climate Research Programme, aiming to coordinate international regional climate downscaling experiments. CORDEX sets a number of directives, including predefined resolution, regions, output variables and formats, to facilitate analysis of possible future climate changes (Nikulin et al., 2012).

Among the different CORDEX regions, Central Asia represents one of the largest domains, covering parts of Europe, Africa and almost the entire Asian continent. The domain extends from eastern Europe to the eastern part of China and from the northern part of India and the Arabian Peninsula in the South, to Siberia and the Arctic ocean (Barents sea and Kara sea) in the North. It includes, almost entirely, two of the most important and populated countries of the World: China and Russia. The region, despite being mainly characterized by arid and semi-arid climatic conditions, presents a wide and differing variety of climatic zones, going from the desertic zones of Gobi and the Arabian peninsula, to the cold and dry areas of Siberia and the wet Northern Indian monsoon region (Ozturk et al., 2017). Therefore, it offers the unique opportunity to test the model sensitivity to different climatic conditions at once.

Beside its importance from a modeling perspective, the extension, geography and the presence of several under-developed or developing economies, makes the CORDEX Central Asia domain one of the most vulnerable regions on Earth to the effects of climate changes. Even small changes in climate conditions could dramatically affect ecosystems, agricultural crops, water resources, human health and livelihood of the region (Siegfried et al., 2012; Lioubimtseva et al., 2005; Lioubimtseva and Henebry, 2009; Liu et al., 2013; Yong-Jian et al., 2013; Zhen-Feng et al., 2013; Wang et al., 2017; Chuluunkhuyag, 2008; Diniega, 2012; Macias-Fauria et al., 2012). In this context, reliable information on potential future changes with high spatial resolution acquire significant importance for the development of effective adaptation and mitigation strategies.

The recognized prerequisite that every climate model has to satisfy, in order to provide reliable future climate projections, is the ability of realistically simulating present-day climate (Kim et al., 2014; Nature-Editorial-Board, 2010; Kim and Lee, 2003). Assessing the ability of a climate model to simulate the current climate is defined as model evaluation (Airey and Hulme, 1995). Model evaluation consists in an assessment of model quality and deficiencies originating from different modeling assumptions, conducted through the comparison of model outputs and observations (Kim et al., 2014; Kim and Lee, 2003; Flato et al., 2013; Lenderink, 2010; Overpeck et al., 2011; Bellprat et al., 2012a, b). Evaluation experiments normally consist in a set of present-days simulations conducted in a perfect boundary setting, i.e., using reanalysis products as lateral boundary forcings. This "modus operandi" allows for the separation of possible model biases from biases due to erroneous large-scale forcings,

thus highlighting specific model deficiencies (Kotlarski et al., 2014). These may be related to the model formulation and to choices in model configuration (Awan et al., 2011; Bellprat et al., 2012a, b; de Elía et al., 2008; Evans et al., 2012). In the second case, it should be possible to improve model performances by testing different model configuration setups and choosing the one that better agrees with observations. This approach might be conceived as an optimization step. Nevertheless,
it is important to emphasize the fact that a specific model configuration could produce better results, by simply compensating for some deficiencies in the model formulation (Hourdin et al., 2017).

In climate models, the complexity and small spatial scales of the physical processes involved, requires the so-called parameterization of many of these processes: this basically consists in summarizing physical phenomena and their interaction across different spatial and temporal scales (Fernández et al., 2007; Rummukainen, 2010; McFarlane, 2011; Hourdin et al., 2017),
which is associated with substantial uncertainties. The same processes may be described through different parameterizations, with a different degree of complexity and infinitive parameter values. Consequently, the outcomes of a climate model might largely differ, depending on the parameterizations used and the selected parameters inputs. Additionally, the use of different forcings datasets, for example for greenhouse gases, aerosols or land cover changes, might have a significant effect on the results. Further, other details that need to be considered when configuring a climate model simulation for a defined domain
are the configuration and spatial resolution of the model grid (both horizontally and vertically) and the coupling with different models representative of other components of the climate system. For regional climate models, all these aspects are domain dependent (Jacob et al., 2007, 2012; Rockel et al., 2008). This means that a regional climate modeler should always evaluate different model configurations, isolating the one that leads to a better agreement with observations, for each investigated region and employed model. In doing so, several sources of uncertainties should be taken into consideration: the fact that performances of the RCM for a specific region might vary according to the boundary conditions, the model internal variability and observational datasets should be acknowledged when evaluating model performances.

RCMs evaluation and configuration have been the subject of a large number of studies, for different regions, such as in Kotlarski et al. (2014); Umakanth and Kesarkar (2018); Borge et al. (2008); García-Díez et al. (2013); Crétat et al. (2012); Rajeevan et al. (2010); Diro et al. (2012); Giorgi et al. (2012); Reboita et al. (2014); Li et al. (2018); Huang et al. (2015).
For the RCM considered in this study, the COSMO-CLM (Rockel et al., 2008), Europe has received larger attention. Bellprat et al. (2012b) applied a quadratic metamodel on a subsample of model parameters in order to objectively tune the model for the region. Their method is considered the reference for COSMO-CLM for determining optimal parameters values and has been further developed and applied to the Mediterranean region by Avgoustoglou et al. (2017) and for higher resolution for the Alpine region by Voudouri et al. (2018). Bellprat et al. (2016) additionally used the same method for the European
and the North American domain, finding quite similar values of optimal model parameters for the two regions. Using a more subjective approach, Montesarchio et al. (2012) conducted a set of sensitivity studies in order to determine the best model setup for the simulation of near surface temperature and precipitation over Northern and Central Italy, at a spatial resolution of $\sim 8$km. Despite several important findings, they were not able to determine an optimal model configuration with respect to these variables. The alpine region was the subject of a study with COSMO-CLM by Suklitsch et al. (2008), where they found
that changes in model resolution has a larger impact on model simulation than modifying dynamical and numerical schemes.

Bachner et al. (2008) conducted a similar work for Germany, focusing on model performances for summer precipitation, concluding that the model uncertainty due to the modified physical parameterizations is considerable and highlighting the need of conducting evaluation and sensitivity studies prior to the application of a model for climate change projections. Studies investigating COSMO-CLM sensitivity for regions out of Europe are more rare. Lange et al. (2015) tested model performances to different convection and non-precipitating subgrid-scale clouds parameterizations for South America. Through this work they managed to reduce long-standing model biases in precipitation for the region, by using the Integrated Forecasting System (IFS (Bechtold et al., 2008)) of the European Centre for Medium-Range Weather Forecasts (ECMWF) convection scheme and statistical schemes for subgrid-scale clouds. Bucchignani et al. (2016) compared the performances of twenty-six different model configurations for the Middle East-North Africa (MENA) CORDEX domain. They found that the model is particularly sensitive for the region to changes in physical and tuning parameters. In particular, they obtained best model performances with the representation of the albedo based on the Moderate Resolution Imaging Spectroradiometer (MODIS (Lawrence and Chase, 2007)) data, and a parameterization of aerosol based on the NASA-GISS Aerosol Optical Depth distributions (Tegen et al., 1997). Finally, Bucchignani et al. (2012) evaluated several configurations for North-Western China at a spatial resolution of approximately $\sim 8$km, even though they did not propose any optimal configuration for the region.

So far, neither an evaluation nor a sensitivity study on the impact of different configurations of an RCM have been documented for the CORDEX Central Asia domain. Such analyses are required to guide further model development and applications for the region: if we want to produce future climate projections for the region, we need to investigate model performances and deficiencies for the area and propose optimal model configurations.

In the light of the upcoming phase of the CORDEX initiative, denominated CORDEX - Coordinated Output for Regional Evaluations (CORE - Gutowski Jr et al., 2016), in this paper the results of a set of sensitivity experiments with the regional climate model COSMO-CLM version 5.0, for the Central Asia CORDEX domain, are presented. In this perspective, this work represents the first step for the production of climate projections for the Central Asia domain using COSMO-CLM, evaluating general model performances, isolating the effects of different uncertainty sources on model results and determining an optimal model configuration for a region for which almost no reference exists. Starting from a reference model setup, general model performances are evaluated, testing the effects of a set of singular physical options and their mutual interaction as well as two different forcing datasets on the simulation of monthly and seasonal values of three variables that are important for impact studies. These are near surface temperature (T2M), precipitation (PRE) and diurnal temperature range (DTR), the latter representing the daily excursion between maximum and minimum temperature, which is particularly important in terms of human body adaptability and stress. The final goal of this study is two-fold: having a general overview of model performances and its uncertainties for the considered region and determining at the same time a "best" suitable model configuration.

In section 2 of this paper, the model, the different datasets and the methods employed in this study are described. Then, in section 3, results are presented. Finally, conclusions are outlined, with a general discussion of model performances and the proposal of a final optimal model configuration for the area of study.

## 2 Methods

In this section the research methods are described, including details on the model and the different simulation setups, the observational datasets used for the evaluation of model results and the employed metrics.

### 2.1 Model and Experiments Description

The Consortium for Small-Scale Modeling in Climate Mode (COSMO-CLM) is a non-hydrostatic regional climate model developed by the CLM-community, an open international network of scientists. The model version employed in this study is the COSMO-CLM 5.0_clm9. Many studies have been conducted in the recent years over different CORDEX regions, using the COSMO-CLM (Panitz et al., 2014; Dobler and Ahrens, 2010; Bucchignani et al., 2016; Smiatek et al., 2016; Jacob et al., 2014; Kotlarski et al., 2014; Zhou et al., 2016).

The simulations presented in this study are performed with a spatial resolution of 0.22°, as specified in the new CORDEX-CORE directives (Gutowski Jr et al., 2016), on a rotated geographical grid. The initial simulation domain extends from ∼3° to ∼145° over longitudes and from ∼16° to ∼73° over latitudes. The domain includes a model relaxation zone of ∼250 km on each domain side, used to "relax" the model variables towards the driving data (Køltzow, 2012; Davies, 1976). Results of the simulation for this area are excluded from the analysis, with the final domain extent shown in Fig. 1. If not differently specified,

all the simulations are run over a fifteen year-long period from 1991 to 2005, with the first five years excluded from the analysis and considered as spinup time.

In a set of sensitivity experiments labeled from **a** to **q** in the first section of Tab. 1, the effects on model performances of different changes in the model configuration are tested, first individually and then combining them with each others. The setup of experiment **a** is the reference from which the other experiments are configured, by implementing the modifications specified

in the table. The model configuration used for the reference simulation is the same used for the CORDEX East Asia domain for the COSMO-CLM model version 5.0. This was considered as a good reference for the purposes of this study, since the two regions share a large part of their domains. A general description of the setup of the reference simulation is provided in Tab. 2.

All the performed simulations are driven by the NCEP version 2 reanalysis data (Kanamitsu et al., 2002), provided as boundary and initial conditions. The boundaries have a temporal resolution of 6 hours and a spectral resolution of T62 (∼

1.89°). NCEP2 data have been selected as boundary data, instead of commonly employed ERAInterim reanalyses (Dee et al., 2011), since their spatial resolution is closer to the one of the three Global Circulation Models (GCMs) that are used for CORDEX-CORE simulations in the CLM community: MPI-ESM (Giorgetta et al., 2013), HadGEM (The HadGEM2 Development Team: Martin et al., 2011) and NorESM (Bentsen et al., 2013; Iversen et al., 2013), with a spatial resolution of, respectively, ∼ 210 km, ∼ 210 × 140 km and ∼ 270 × 210 km. Thus, using NCEP2 as drivers allows to reproduce a resolution

jump more similar to the one present when using the considered GCMs.

Acknowledging the fact that ERAInterim reanalysis data, which have a spectral resolution of T255 (∼ 0.7°), are normally employed for the evaluation of RCMs, two additional simulations are performed, driven by ERAInterim (second section of

Tab. 1). This allows to estimate the effects of the two different driving data on the simulations results and to support possible conclusions on an optimal setup, verifying how significantly the results differ in the two cases.

In order to better discriminate different sources of uncertainties in the model simulations, a run covering the period 1991-2005 is also performed (third section of Tab. 1), using a timestep of 120 s, instead of the one of the reference simulation of 150 s.

Two 25-year long simulations, covering the period 1991-2015, are performed for testing different configurations that could help in reducing model biases over areas characterized by the presence of permafrost in winter. The two simulations, labeled **SOIL** and **SNOW** in the fourth section of Tab. 1, are performed increasing in both the number of soil layers from 10 to 13, together with their total depth from approximately 15 m to more than 130 m, and, only for SNOW, additionally using the multi-layer snow model of COSMO-CLM (Machulskaya, 2015). These simulations cover a longer period than the others, since a longer spinup time is necessary in order to account for more and deeper soil layers. Their results, excluded from the direct comparison with the other simulations, are discussed in the results and concluding sections of this paper.

Finally, a set of four simulations are additionally performed for the investigation of the model internal variability (last section of Tab. 1). These simulations have the same setup as the reference simulation **a**, but are initialized at four different dates, shifted by +/- 1 and 3 months with respect to the reference one.

All the proposed simulations are designed with the goal of better understanding main model limitations for the area and to which degree they can be reduced by properly configuring the model, isolating the effects of different sources of uncertainties.

## 2.2 Observations

Gridded observational datasets are used to compare model results against observational data on a similar scale. These gridded datasets are obtained through statistical extrapolations of surface observations. In addition to uncertainties related to the original measurements, these datasets also contain important uncertainties due to the statistical extrapolation procedure (Flaounas et al., 2012; Gómez-Navarro et al., 2012). For climate model evaluation studies, these uncertainties are usually taken into account by using a range of different datasets (Collins et al., 2013; Gómez-Navarro et al., 2012; Bellprat et al., 2012a, b; Flaounas et al., 2012; Lange et al., 2015; Zhou et al., 2016; Solman et al., 2013).

In this study, the issue of observational uncertainties is addressed by considering three different datasets for each of the investigated variables. The datasets include both observations and reanalysis data. For all the three considered variables, information is retrieved from the CRU TS4.1 observational dataset (Harris and Jones, 2017). Information on T2M and PRE is also retrieved from the University of Delaware (UDel) gridded dataset (Willmott, 2000), provided by the NOAA/OAR/ESRL PSD, Boulder, Colorado, USA. For T2M and DTR, in addition, the Modern-Era Retrospective analysis for Research and Applications, version 2 (MERRA2) (Gelaro et al., 2017) is employed. For precipitation, the third considered dataset is the Global Precipitation Climatology Centre dataset (GPCC) (Becker et al., 2011), while the ERAInterim reanalysis dataset (Dee et al., 2011) is used in addition to MERRA2 and to CRU for the evaluation of DTR.

All the datasets are retrieved on a grid with the same spatial resolution of 0.5. The ERAInterim data, that originally have a horizontal resolution of approximately 80km, are interpolated to the same grid resolution. The output of the simulations is

upscaled to the same 0.5° grid of the observations. For temperature and diurnal temperature range, a bilinear remapping method is used for the upscaling, while for precipitation a conservative remapping method is employed.

Fig. 2 shows the spread of the different observational datasets for each variable, for yearly, winter and summer climatological means over the period 1996-2005. As evident, large differences emerge among the different datasets, in particular for regions characterized by complex topography and lower observational stations density, such as the Tibetan Plateau and Siberia. The given spread could make it hard to quantify model biases over certain regions. In the case of T2M and DTR, the spread is certainly influenced by the fact that some of the datasets are reanalyses. Nevertheless, for T2M, differences exceeding 8° are present, in particular in winter, even between the CRU and the UDEL, over regions where the interpolation is highly affected by the low number of stations (Matsuura and Wilmott, 2012; Bucchignani et al., 2014). For PRE, the spread in the different observations (expressed in percentage with respect to the GPCC values) is remarkable in winter over the Tibetan Plateau and in summer over particularly dry areas. Despite the differences might likely be influenced by the employed interpolation methods in each case, the spatial coverage of observation is still considered their main driver (Dong and Sun, 2018; Matsuura and Wilmott, 2012; Sun et al., 2018; Naumann et al., 2014).

## 2.3 Analysis Details and Evaluation Metrics

In order to rank different model configurations according to their skills in simulating the three considered variables over the region, their performances are evaluated with respect to the ones of the reference simulation (**a**, Tab. 2).

Since in the context of CORDEX simulations the main interest is often on the comparison of the mean climate between two different periods in time, the primary focus of the proposed analyses is on climatological monthly values of the considered variables: monthly values of daily means of T2M and DTR are considered, while for PRE integrated values are used. In addition, the results are supported by the investigation of the simulated monthly variability.

In the latter case, since the model is not expected to exactly match the observed temporal evolution of the investigated variables point by point (Gleckler et al., 2008; Wilks, 2006), regional mean anomalies are considered. For each grid point in the domain, monthly anomalies are first calculated by subtracting the climatological mean from each monthly value. The variability is then analyzed based on these anomalies averaged over sub-regions characterized by similar climate conditions.

The decomposition of the domain into a set of sub-regions is obtained by means of a k-means clustering (Steinhaus, 1956; Ball and Hall Dj, 1965; MacQueen et al., 1967; Lloyd, 1982; Jain, 2010) of quantile-normalized (q-normalized) monthly climatologies of the investigated variables. K-means is a clustering technique using the concept of Euclidean distance from the centroids of a pre-determined group of clusters, for separating similar data into groups. For the purposes of this paper, following several tests and the results of other studies (Mannig et al., 2013), a total number of 11 clusters is selected. The k-means clustering algorithm is reiterated over 3000 times in order to achieve the presented results, using q-normalized values of monthly climatologies of T2M and DTR derived from the CRU dataset and PRE values derived from the GPCC as input. Fig. 3 shows the results of the k-means clustering. The mean climatologies of the considered regions for the three investigated variables are also reported in Tab. 3.

For both the analyses of mean climate and variability, metrics adapted from Gleckler et al. (2008) are used. In the following subsections, we give more details on the employed metrics.

### 2.3.1 Climatology

For the evaluation of the climatological means, we employ a Skill Score (SS) metrics expressed as:

$$SS = (1 - \frac{(MAE)_{exp}}{(MAE)_{ref}}) \times 100 \qquad (1)$$

where the Mean Absolute Error (MAE) is given by:

$$MAE = \frac{1}{W} \sum_{i=1} \sum_{j=1} \sum_{m=1} w_{ijm}|sim_{ijm} - obs_{ijm}| \qquad (2)$$

where *sim* and *obs* are the monthly climatological means of, respectively, the considered simulation and observational dataset. The indices *i*, *j* and *m* vary, respectively, over longitudes, latitudes and months of a year. *W* is the sum of the weights $w_{ijm}$, taking into account the different lengths of the months and the grid boxes effective area. The *SS* is calculated with respect to a reference simulation. Positive values indicate an improvement of the considered simulation *exp* with respect to the reference *ref*, while negative values indicate worse performances. The analyses of MAE for the mean seasonal cycle are conducted for the entire domain and single sub-regions. Additionally, the same metrics are applied for the analysis of single seasons for the entire area.

### 2.3.2 Variability

The analysis of the model performances in simulating the mean climate is complemented by the investigation of simulated variability.

There is no reason to expect models and observations to agree on the phasing of internal (unforced) variations. Hence metrics such as MAE are not appropriate for characterizing the model performances for interannual variability (Gleckler et al., 2008). Here, for an overall evaluation of the simulated variance in the different cases, the ratio of simulated to observed variance is considered:

$$Variance\ ratio = \frac{\sigma_{exp}^2}{\sigma_{obs}^2} \qquad (3)$$

It is important to mention that correctly matching the observed variance does not guarantee a correct representation of the modes of variability associated with this variance.

For taking into account observational uncertainties, all the proposed analyses are conducted separately for each observational dataset. Changes in model performances for a given configuration are considered relevant only when consistent among the different observations.

## 3 Results

In this section the results of the conducted analyses are presented, starting from the consideration of climatological means and followed by the analyses of simulated-to-observed variability.

### 3.1 Mean Climate

In order to characterize the general performances of the model over the region, for the three considered variables, maps of the yearly, winter and summer mean biases of the reference model simulation **a** with respect to the different observational datasets, are first presented.

Fig. 4 shows that for T2M, the largest biases are evident in winter (central panels), with warmer simulated conditions over the northeastern part of the domain, where the bias in some case exceeds 15°C. These exaggerated biases are mainly relative to the UDEL dataset and, in general, particularly large biases are limited to a few points characterized by complex topography and lower stations density, where the gridded datasets are less reliable. When the CRU dataset is considered, the values of the bias rarely exceed (are below) 10°C (-10°C), for really few points. Beside these points, still some remarkable biases are present but these are well within the ranges of other CORDEX simulations for the area (Wang et al., 2013; Bucchignani et al., 2014; Ozturk et al., 2012). In summer (Fig. 4, right panels), a positive bias (ranging from +5°C to +10°C) is present over the central and south-western part of the domain, in arid and desertic areas such as the Arabian Peninsula and the Taklamakan desert. Conversely, a cold bias is present over Siberia in this case, with values rarely below -5°C. Biases of annual mean values (Fig. 4, left panels) are smaller than the seasonal ones, except for the Tibetan Plateau. Here a similar particularly pronounced cold bias is evident, for all seasons, with respect to all observational datasets, with values sometimes smaller than -10°C. In this case the observations are likely less reliable. In general, the simulation results are in better agreement with the MERRA2 dataset than with the CRU and the UDEL. Nevertheless, despite the evinced differences in the magnitude of the model biases against different observational datasets, their spatial patterns are very similar in all the cases.

Concerning PRE (Fig. 5), remarkable biases are present in the winter and summer as well as in the annual mean for all the observational datasets. The biases in this case are expressed as percentage with respect to the values of the corresponding observational estimates. In summer (Fig. 5, right panels), a particularly pronounced negative bias, with values down to -100%, is visible over arid regions and the monsoon area. This is of the same order of the spread of the observational datasets for the area (Fig. 2). Over the Tibetan Plateau the bias in summer is positive, with values larger than 100%. In winter (Fig. 5, central panels), this positive bias becomes even larger (but again in the order of the spread of observations), and extends further over adjacent regions. Over the central part of the domain, a different behavior is evident between winter and summer: while in winter the model simulates wetter conditions (+20% to +100%), summers are drier (∼ -50 %) than in observations. In the

annual mean (Fig. 5, left panels), the simulated climate is wetter over a large part of the eastern domain (with values exceeding +100%) and drier over desert zones (with rare values smaller than -80%). In this case, over the central part of the domain, winter and summer biases compensate each other.

In all the cases, the simulated DTRs are smaller than the observed ones over almost the entire domain (Fig. 6). A positive bias in DTR, rarely exceeding +5°C, is evident only over isolated parts of the southern domain, in particular over the southern borders of the Tibetan Plateau. The differences arising from the comparison against CRU observations are more pronounced than the ones against reanalysis data, with biases lower than -10°C in some cases. The pattern of the bias is quite similar for all the three considered datasets, with some larger differences in summer. Over the northernmost part of the domain, characterized by particularly cold conditions (minimum temperature under -30°C in winter, see Tab. 3), a strong negative bias is evident only with respect to the CRU in all seasons. The smaller bias over this area arising from the comparison against reanalysis data is most likely due to the nature of these datasets, which combine model predictions and observations.

The additional simulations performed with the same reference setup but with a different timestep (**TIMESTEP**, Tab.1) and driven by ERAInterim instead of NCEP2 (**a_ERAInterim**, Tab.1), lead to very similar biases, for all variables (see supplements). This suggests that evinced biases are likely inherent to the model formulation itself.

### 3.1.1 SS - Seasonal Cycle

In this section, the results of the Skill Score (SS) derived from the MAE calculated over the mean seasonal cycle and all the points of the domain are presented.

Fig. 7 (upper row) shows that for T2M, among the experiments for which single changes are applied to the reference model configuration (left side of the dotted vertical line), the ones with changes in the albedo treatment (**c+d**) lead to a noticeable improvement of the results (ranging between +4.5% and +7%). Nevertheless, in this case, the largest improvements (greater than 5% for all the observational datasets) are obtained for experiment **j**, in which the type of the hydraulic lower boundary accounts for ground water with drainage and diffusion. Combining the configuration changes of different experiments (right side of Fig. 7 ) the results for temperature are considerably improved, whenever either one of the setup changes of experiments **d** or **j** are used, with values of SS larger than 4% in almost all the cases. Other "combined" experiments do not have an important effect on the results.

For PRE (middle row in Fig. 7), only the results of one experiment, among the ones with single changes in the model configuration, are improved compared to the reference: experiment **d** (SS=∼+4%), in which the albedo is modified considering the forest fraction. The positive effect of this change is slightly enhanced when used jointly with other configuration choices (experiments **m,n,o,p,q**), having indeed an important effect on precipitation.

As for PRE, also for DTR (Fig. 7, bottom row) only one experiment seems to sensibly improve over the results of the reference simulation: experiment **i** (SS ranging between +4% and +5%). In this experiment, the soil heat conductivity takes into account the ratio of soil moisture to soil ice. For DTR two experiments, **d** and **j**, lead to particularly negative skills (SS between -4% and -5%), which also affect the combined experiments including their configuration changes. The unique

exception is the combined experiment **q**: in this case, the negative effects on the simulation of DTR of experiment **d**, seem to be compensated by the positive ones of experiment **i**, resulting in positive values of SS, varying between +1% and +2%.

In summary, the presented analyses show that the combined representation of surface albedo (taking into account forest fraction) and soil heat conductivity (accounting for the ratio between ice and moisture in the soil), as configured in experiment **q**, has the best positive effects on the representation of the mean seasonal cycle of all the three considered variables, among all the tested configurations.

Although some differences in the results of the SS calculated based on the different observational datasets are evident, experiment **q** shows the same positive sign of improvement in all the cases. This is also true when comparing the results obtained driving the same simulations **q** and **a** with NCEP2 and ERA-Interim reanalysis data (Tab. 4). This confirms the potential of experiment **q** in improving model performances for the area.

### 3.1.2 SS - Sub-domains

The same SS analyses for the mean seasonal cycle are conducted for sub-regions characterized by similar climate conditions. This allows to test the model sensitivity for regions where different physical processes might play a different role. The analyses presented in Fig. 8 are conducted, as in the case of the entire domain, separately for different observational datasets. Here, for visualization reasons, the magnitude of changes in the SS is only reported for a reference dataset for each variable, being the CRU for T2M and DTR and the GPCC for PRE. At the same time, for each experiment and subregion where the sign of changes in SS is the same among the different observations, a point is drawn.

Fig. 8 basically confirms the results of the SS calculated for the entire domain. For the experiments with single changes in the reference model configuration it is possible to see that for T2M (upper panel), the most relevant SS changes are obtained for experiment **d** and **j**, with improvements exceeding 30% over some region. For PRE (middle panel), improvements over all the clusters are obtained only for experiment **d**, and positive SS values are evident only for few other experiments for specific sub-domains. In this case, changes in SS are smaller than for T2M, rarely exceeding 10%. For DTR, experiment **i** allows to achieve improvement in model performances up to 25% with respect to the reference simulation, not visible in the other cases, for almost all the sub-domains.

Among the combined experiments (right side of each panel of Fig. 8), it is possible to affirm that experiments **m** and **q** present similar performances for T2M and PRE. Conversely, only experiment **q** shows an important improvement in model performances on more than one subdomain for DTR .

Fig. 8 shows that it is almost impossible to achieve an improvement in model performances for all regions and variables. Despite experiment **q** presents positive SS values for a large majority of sub-domains, some negative values are also evident for specific sub-regions. This also happens when considering different variables. For example, for experiment **d**, improvements over the entire domain are evident for T2M and PRE, while the same setup leads to worse model performances in terms of DTR, for almost the entire domain. Therefore, even though important improvements are obtained in different cases, it is crucial to highlight the fact that these might be the product of some compensation effect over different variables and domain sub-regions.

### 3.1.3 SS - Single Seasons

The same SS analyses are additionally conducted for the monthly climatologies of each season, over the entire domain. Seasonal analyses could help in discriminating simulations presenting good and coherent performances over more periods of the year. The results, reported in the supplementary section of this paper, show that the largest changes in the seasonal values of
SS are obtained for summer, for processes related to the representation of surface and soil properties. On the other hand, in winter the changes in model performances among the different experiments are substantially smaller than in the other seasons. Overall, for single seasons, the most important and consistent improvements in the simulated climatological mean of the considered variables with respect to the reference simulation are obtained for experiment **q**, confirming the results obtained for the seasonal cycle.

### 3.1.4 Effects of Soil Depth and Snow Model on mean winter temperatures

The two simulations (**SOIL** and **SNOW**) specifically designed for testing the effects of changes in soil depth and the use of a multi-layer snow model on the COSMO-CLM simulation of T2M over areas characterized by the presence of permafrost and snow in winter, do not present significantly different results than the reference simulation (Fig. 9). In particular, they do not allow to reduce the warm bias in T2M present for the reference simulation in winter over the Western Siberia part of the
domain, with even warmer conditions simulated by experiment **SNOW** (Fig. 9, left).

### 3.2 Variability

In this section, the results of the analysis of simulated variance is presented, with the goal of complementing the analyses of the mean climate of Sec. 3.1. First, a general overview of the model skill in simulating observational variability is described, followed by a discussion of the different uncertainties affecting this metrics.
Fig. 10 shows the ratio of variance of the different COSMO-CLM experiments with respect to the one of the observations. The variances are calculated from monthly anomalies values of the three considered variables averaged over the subregions shown in Fig. 3 (see also Sec. 2.3). For visualization reasons, a single observational dataset is used for each of the considered variables in this case: CRU for T2M and DTR, and GPCC for PRE. The results of the comparison against other observational datasets are considered when discussing different sources of uncertainties in Sec. 3.2.1.
In general, for DTR and T2M, there are no large differences in the variance ratios of all the experiments, except for a few sub-regions. For PRE, conditions are more heterogeneous, with relatively large differences among all the simulations. Nevertheless, the most pronounced changes are still limited to a few clusters.

For T2M, the best results in terms of simulated variance are obtained: the model is able to reproduce the interannual variability of the observations particularly well. In particular, a good agreement between simulated data and observations is evident
for subregions **WSC**, **IMO** and **ARC**. The largest underestimation of the observed variance of T2M is obtained for cluster **CSA**. Therefore, the model is not only unable to simulate the mean temperatures of particularly cold areas, as demonstrated in Sec. 3.1, but it also shows a very low variability for the same variable, over these regions, when compared to observations.

A particulalry small value of the variance ratio is also evident for T2M for the sub-domains **DSS**, **SAR** and **STE** throughout almost all the experiments. These regions are all characterized by a large range between minimum and maximum monthly temperatures (see Tab. 3).

For PRE, in general, the values of the ratio of simulated-to-observed variance are considerably larger than 1 for almost all the experiments and subdomains. Values are closer to 1 only for the domains **WSC** and **DHS** throughout all the experiments. In the domains **CSA**, **DSS** and **TIB**, variance ratios are particularly remarkable, reaching a value of 3 for some experiment. Over these domains, characterized by complex topography, results from Sec. 3.1 have shown that the model simulates significantly wetter conditions. Hence, for mountainous areas of the domain the model overestimates both mean values and variability of precipitation.

Values of variance ratios for DTR are smaller than 1 over almost all the subdomains and simulations. This indicates that the model, beside underestimating climatological values of the observed temperature diurnal cycle over the entire Central Asia domain as demonstrated in Sec. 3.1, also undererestimates the amplitude of variations in the monthly means.

### 3.2.1 Uncertainties in the Investigation of Simulated Variability

In this section, the influence of uncertainties associated with the observational datasets, boundary data and internal variability on the evaluation of simulated variability are quantified. To investigate the effect of the model internal variability, four additional simulations have been conducted using the setup of the reference simulation, but shifting the initial date by +/- 1 and +/-3 months.

Left columns of each panel of Fig. 11 show the absolute differences in the variance ratio of experiment **a** calculated, for each variable, with respect to different observational datasets. In addition, the right columns of the same figure show the absolute differences in the variance ratio between experiment **a** and the other experiments. The range of changes in the two cases is comparable for almost all clusters and variables. In many cases, the changes resulting from the use of different observations are larger than the differences between the experiments. In these cases, the observational uncertainty is thus too large to allow for a classification of the different experiments in terms of their skill in reproducing the observed variance. The influence of the observational datasets on the variance ratios is larger for PRE and DTR than for T2M.

Despite variations in the boundary data and in the simulated internal variability (as quantified in the additional experiments with shifted initial dates) do not have the same strong effect on the simulated variance ratio as the observational uncertainties, for some regions their values are still comparable to the differences between the simulations (not shown).

In conclusion, the fact that different uncertainties are in the same order of magnitude as the differences between the simulations does not allow for a classification of the different experiments with respect to their skill in representing the observed variability.

## 4 Discussion and Conclusions

The main goal of this work is to evaluate a set of different configuration setups of the regional climate model COSMO-CLM over the CORDEX Central Asia domain, and to isolate different sources of uncertainties, in order to quantify general model performances and to provide a basis for possible improvements of the model simulations for this region. The results of this study are of fundamental importance in the light of the next phase of the CORDEX initiative, in particular considering the vulnerability of the region to the possible effects of climate change.

Concerning the simulation of the mean climate, the model shows remarkable deficiencies in simulating the three considered variables (near surface temperature, precipitation and diurnal temperature range) over different areas of Central Asia and different seasons. Even though over specific areas of the domain these biases are hard to be quantitatively assessed, due to high uncertainties in the considered observational datasets, their spatial pattern is similar in all the cases.

For temperature, large positive model biases are present in winter over Siberia, with remarkable values exceeding +10°C in some cases. There are two likely reasons for these biases: an unsatisfactory representation of snow cover and soil permafrost. In fact, both these factors have a significant impact on heat transport within the soil and heat flux between soil and atmosphere, with important effects on near surface temperatures (Frauenfeld et al., 2004; Lachenbruch and Marshall, 1986; Saito et al., 2007; Klehmet, 2014). Siberian permafrost often exceeds a depth of 100 meters, reaching values of up to 1km (Yershov, 2004). Therefore, many studies (Alexeev et al., 2007; Dankers et al., 2011; Nicolsky et al., 2007; Lawrence et al., 2008; Saito et al., 2007; Klehmet, 2014) highlight the importance of an adequate depth of model soil layers for the proper representation of processes related to permafrost. At the same time, other studies (Saito et al., 2007; Waliser et al., 2011; Klehmet, 2014) suggest that a better representation of the vertical stratification of the snow pack could have a significant effect on the simulated energy budget and, consequently, on near surface temperatures over the area. Following these hypotheses, two 25-year long additional simulations have been conducted during this study, with an increase of the total model soil depth and with the use of a multi-layer snow model. Results indicate that, for the part of Siberia included in the domain of study, no significant changes are evident in the two cases. This further demarcates model limitations, pointing to a structural problem in the model formulation and to the need of new parameterizations for the simulation of processes related to snow cover and permafrost in COSMO-CLM. Importantly, an additional cold bias, in some cases lower than -10°C, is present for every season over the Tibetan Plateau. Other regional climate models suffer from a similar bias (Guo et al., 2018; Meng et al., 2018). Acknowledging the fact that for this area the observational uncertainty is particularly high, the evinced biases could partly be related to a bad representation of the albedo for highly complex topographies. In fact, a study by Meng et al. (2018) showed that changes in the albedo over the region have led to an important improvement of the results of an RCM. Another possible explanation for this cold bias might be the parametrization of surface fluxes (Zhuo et al., 2016). Consequently, further analyses should focus on improving the mode representation of these processes.

For precipitation, particularly wet conditions are simulated by the COSMO-CLM over the Tibetan Plateau. This bias seems to be common to several RCMs for areas characterized by complex topography (Guo et al., 2018; Gao et al., 2015; Feng and Fu, 2006) and is likely related to an overestimation of orographic precipitation enhancement in the models (Gerber et al.,

2018) and/or to an incorrect simulation of the planetary boundary layer (Xu et al., 2016). Additionally, in the COSMO-CLM simulations a significant dry bias occurs over arid and desertic regions, especially in summer. A similar COSMO-CLM bias has already been seen for other semi-arid and dry regions of the World, such as the Mediterranean region. In this case, it was connected with an incorrect simulation of soil-atmosphere interactions by the model (Fischer et al., 2007; Seneviratne et al., 2010; Russo and Cubasch, 2016), which is likely the case also for Central Asia. For both the Tibetan Plateau and arid summer areas, it is important to note that the biases are in the same order of the spread of the observations.

The model underestimates the climatological mean of the diurnal cycle of temperatures, for all the seasons and sub-regions of the domain. This bias is relatively homogeneous over the entire domain of study. Several studies have shown that RCMs typically underestimate diurnal temperature ranges over different parts of the World (Kyselỳ and Plavcová, 2012; Mearns et al., 1995; Laprise et al., 2003). The main factors responsible for these deficiencies seem to be errors in the simulation of the atmospheric circulation, cloud cover and heat and moisture fluxes between surface and atmosphere.

The evinced model limitations for the mean climate do not seem to differ significantly when considering ERAInterim as driving data and a different timestep.

In order to test whether it is possible to reduce the determined model biases, and to which degree, sensitivity experiments have been performed to study the effect of different changes in the configuration of COSMO-CLM and their mutual interaction. After considering different sources of uncertainties, the combined change of the albedo taking into consideration the ratio of forest fractions and the soil conductivity taking into account the ratio of liquid water and ice in the soil, leads to the best results in simulated climatological means of the three considered variables (experiment **q**). Importantly, the model seems to be particularly sensitive to those parameterizations that deal with soil and surface features, and that could positively affect the repartition of incoming radiation.

An analysis of model performances in simulating climatological means per sub-regions characterized by similar physical processes shows that different model configurations may lead to improvements exceeding 30% over some region. The analysis for sub-regions is coherent with the SS analyses conducted over the entire domain, with experiment **q** presenting the best performances over the largest majority of regions, for all variables, in the two cases. Nevertheless, sub-regions analyses show that improvements in model performances are not homogeneous among all the sub-regions and variables, but they are the result of some compensation effect in the different cases.

The investigation of model performances for the simulation of seasonal climatologies confirms the results obtained for the seasonal cycle, with experiment **q** leading to the best and most consistent results among all the seasons. For all the analyzed variables, winter is the season for which no substantial improvements in model results can be achieved with the set of investigated configurations. This suggests that other factors, playing a crucial role for the simulation of Central Asia winter climate, are not properly considered in COSMO-CLM.

The model improvements in the simulation of climatological means, with the same optimal configuration, are very similar when considering different observational datasets, and ERAInterim instead of NCEP2 as drivers.

Finally, the observed variability of temperature is relatively well represented in the model simulations for different sub-regions of the domain. For precipitation, the model overestimates the variability of observations. On the contrary, the model

underestimates the variability in the diurnal cycle of temperatures over the entire region. Among the three investigated variables, only for precipitation there are significant changes in the simulated variance throughout all conducted experiments. However, due to the large uncertainties in the variability estimates from observations, the use of different boundaries and the model internal variability, it has not been possible to rank the different simulations according to their representation of the monthly variability.

*Code and data availability.*  All the data upon which this research is based are available through personal communication with the authors. The codes used for the postprocessing of model results are available at the following links:

https://doi.org/10.5281/zenodo.3469945

https://doi.org/10.5281/zenodo.3516622

The model configuration files of the performed sensitivity tests can be consulted at:

https://doi.org/10.5281/zenodo.3516676

The data derived from the model outputs and on which all the presented analyses are derived, can be downloaded at:

https://zenodo.org/badge/DOI/10.5281/zenodo.3516708

*Author contributions.*  The simulations of this research were performed by ER. All the authors equally contributed to the discussion of the results. The paper structure as well as most of the presented experiments were designed by ER and IK. All authors gave a substantial contribute to the revision of the text and to the formatting of the paper.

*Competing interests.*  No competing interests are present in the paper.

*Acknowledgements.*  This study was funded by the Federal Ministry of Education and Research of Germany (BMBF) as part of the CAME II project (Central Asia: Climatic Tipping Points & Their Consequences), project number 03G0863G.

The computational resources necessary for conducting the experiments presented in this research were made available by the German Climate Computing Center (DKRZ).

The authors are also particularly grateful to the CLM community for all their efforts in developing the COSMO-CLM model and making its code available.

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

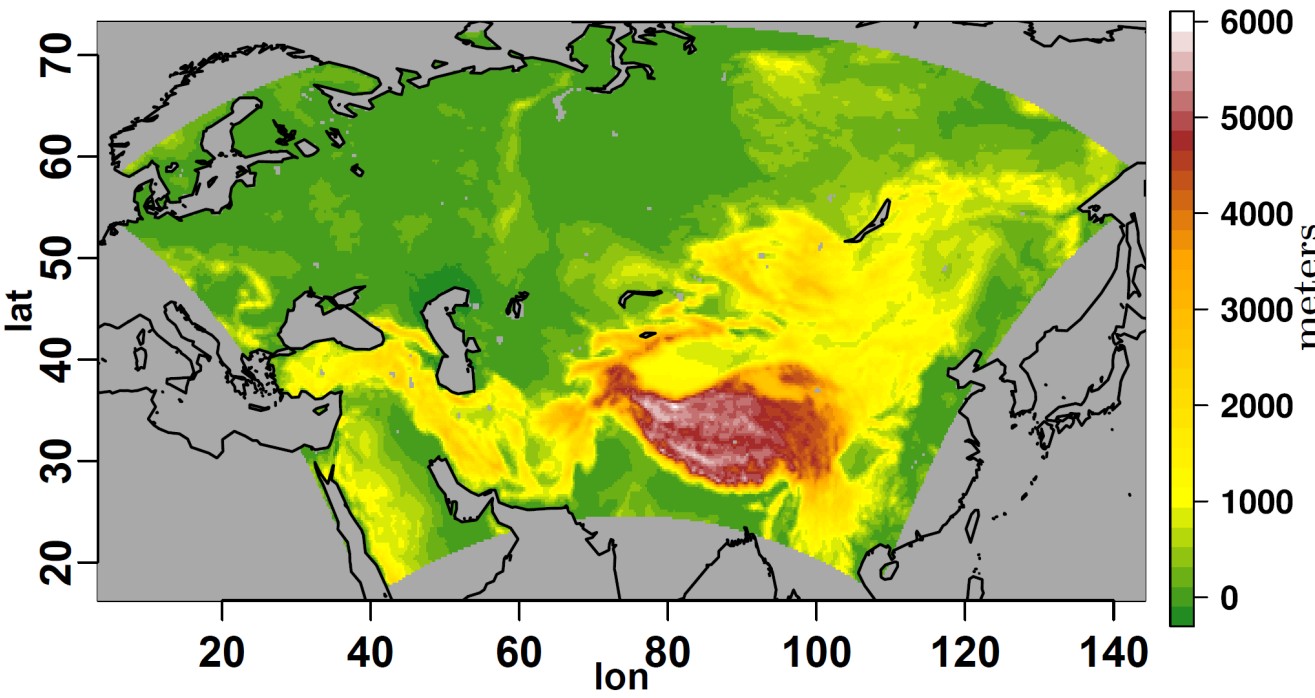

**Figure 1.** Orography map of the Central Asia simulation domain at a spatial resolution of 0.22°. Terrain height information is derived from the Global Land One kilometre Base Elevation (GLOBE - Hastings et al., 1999) dataset.

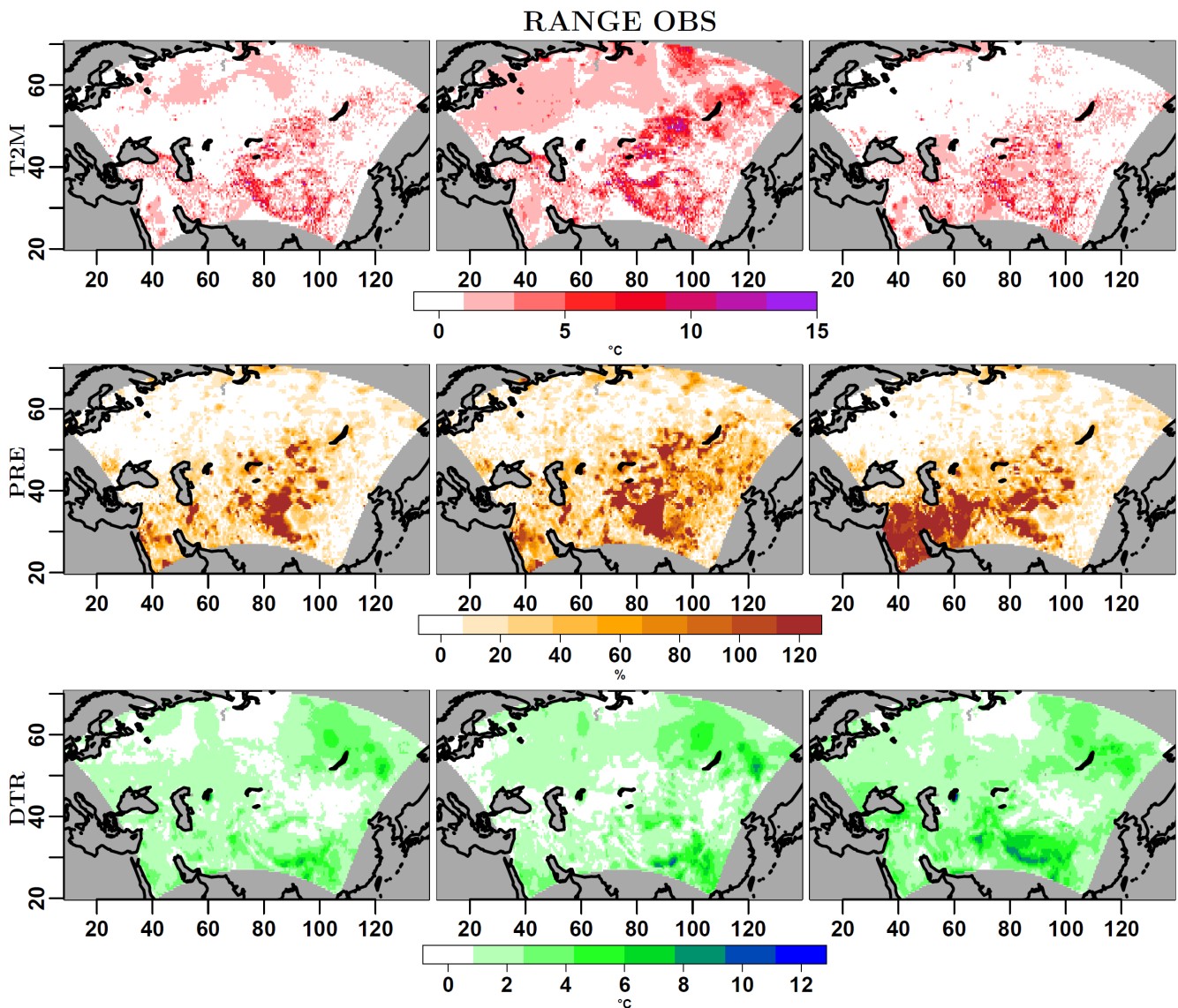

**Figure 2.** Maps of the spread calculated among different observational datasets, for each considered variable, for the annual (*left*), winter (*middle*) and summer means (*right*). From top to bottom, the values for near surface temperature (T2M, °C), relative precipitation (PRE, %) and diurnal temperature range (DTR, °C, bottom panel) are respectively represented.

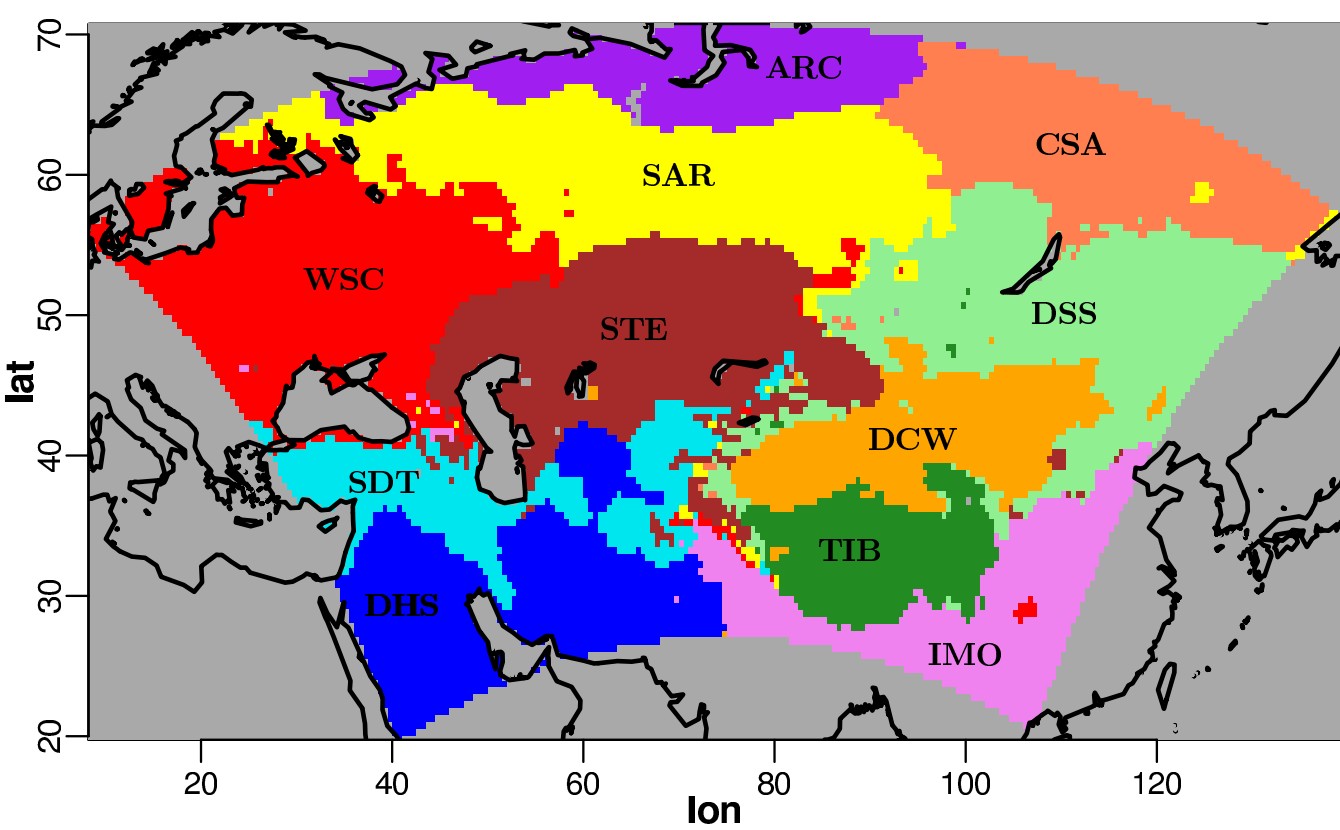

**Figure 3.** Map of the 11 subdomains obtained through k-means clustering of the q-normalized monthly climatologies of the three considered variables over the period 1996-2005.

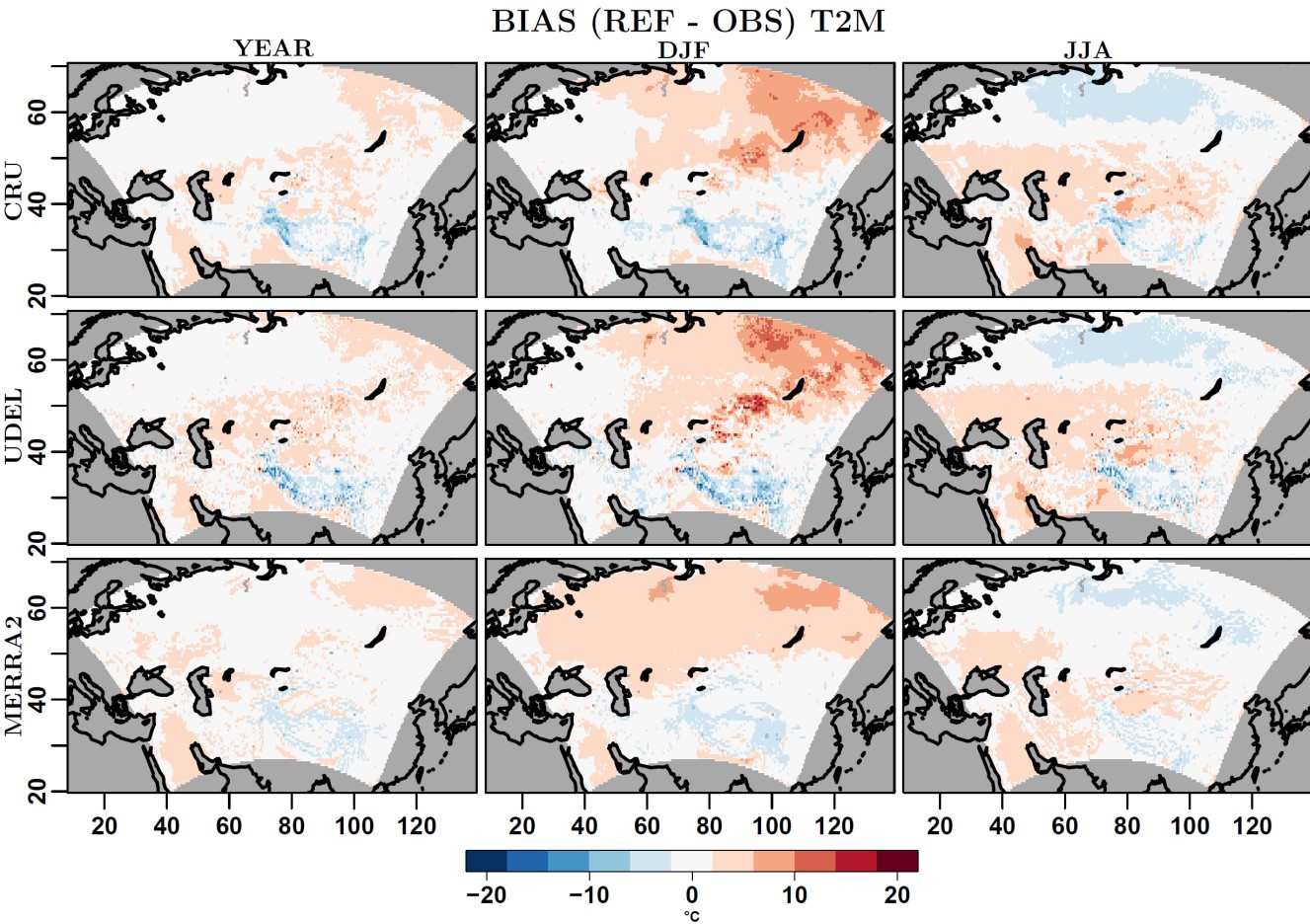

**Figure 4.** Mean bias of annual mean (*left*), winter mean (*middle*) and summer mean (*right*) near surface temperature (T2M,$^{o}$C), of the reference COSMO-CLM simulation (**a**) with respect to three observational datasets (from top to bottom: CRU, UDEL and MERRA2), for the period 1995-2005.

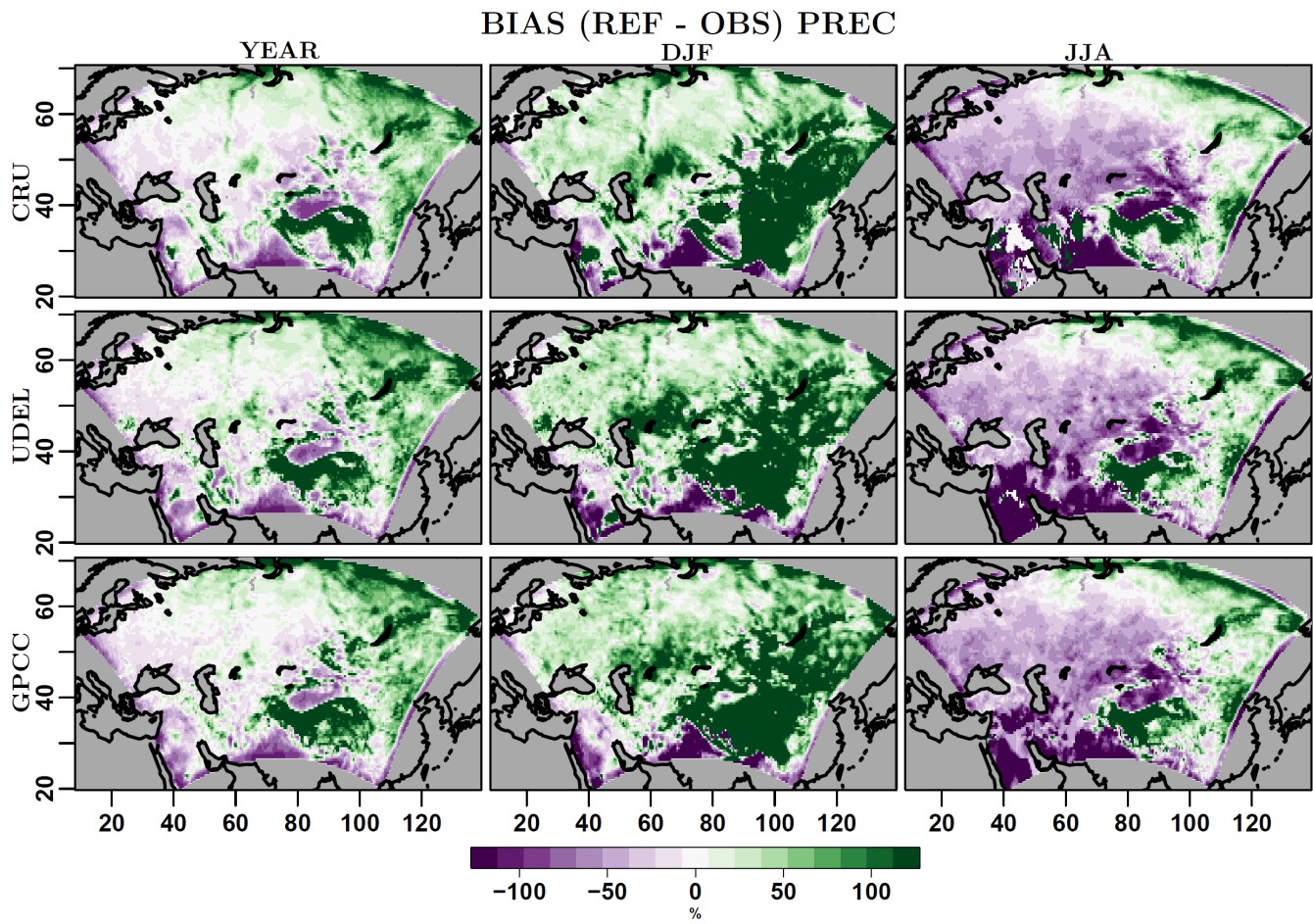

**Figure 5.** Mean bias of annual (*left*), winter (*middle*) and summer mean (*right*) relative precipitation (PRE, %), of the reference COSMO-CLM simulation (**a**) with respect to three observational datasets (from top to bottom: CRU, UDEL and GPCC), for the period 1995-2005.

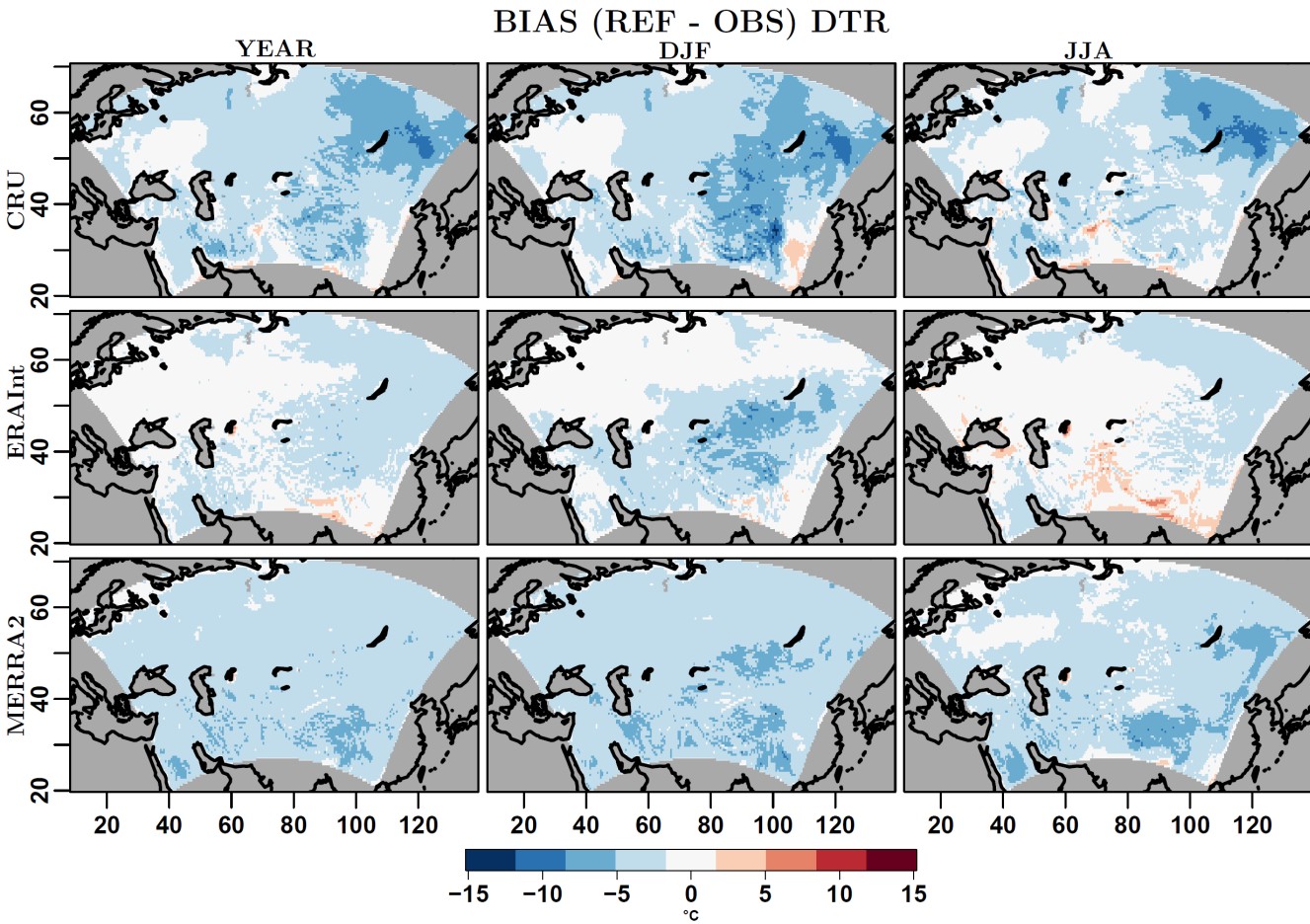

**Figure 6.** Mean bias of annual (*left*), winter (*middle*) and summer mean (*right*) diurnal temperature range (DTR,$^{o}$C), of the reference COSMO-CLM simulation (**a**) with respect to three observational datasets (from top to bottom: CRU, MERRA2 and ERAInterim), for the period 1995-2005.

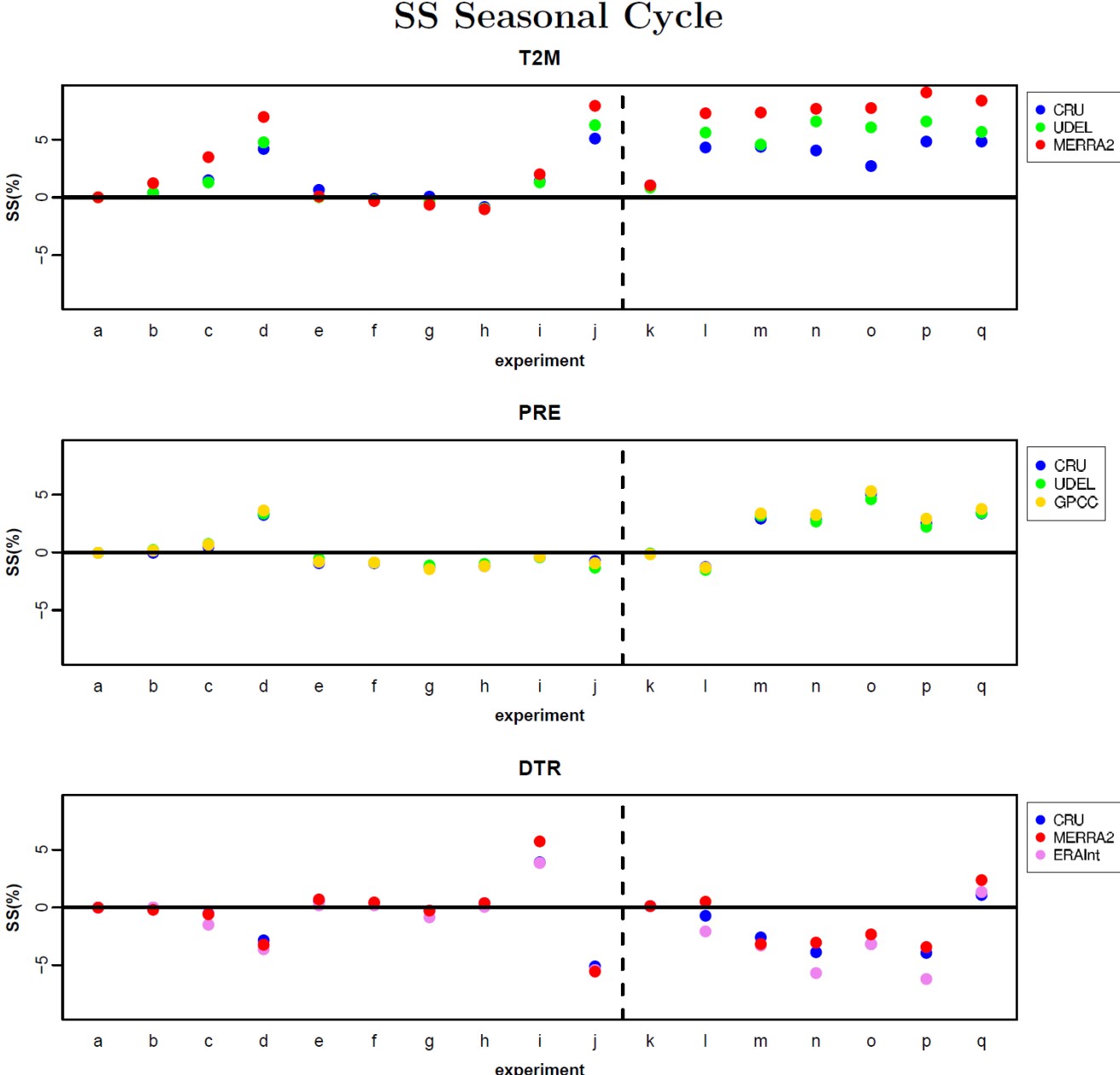

**Figure 7.** Skill Score (SS) derived from the MAE calculated over the monthly climatological values of the seasonal cycle of different COSMO-CLM simulations and observational datasets. From top to bottom, the SS for near surface temperature (T2M), precipitation (PRE) and diurnal temperature range (DTR) is respectively displayed. The dotted vertical black line divides the simulations with the same configuration of the reference simulation plus a single change in the model setup (*left*) and the ones obtained through the combinations of the previous ones (*right*). Positive (negative) values indicate better (worse) performances of the considered simulations compared to the reference one.

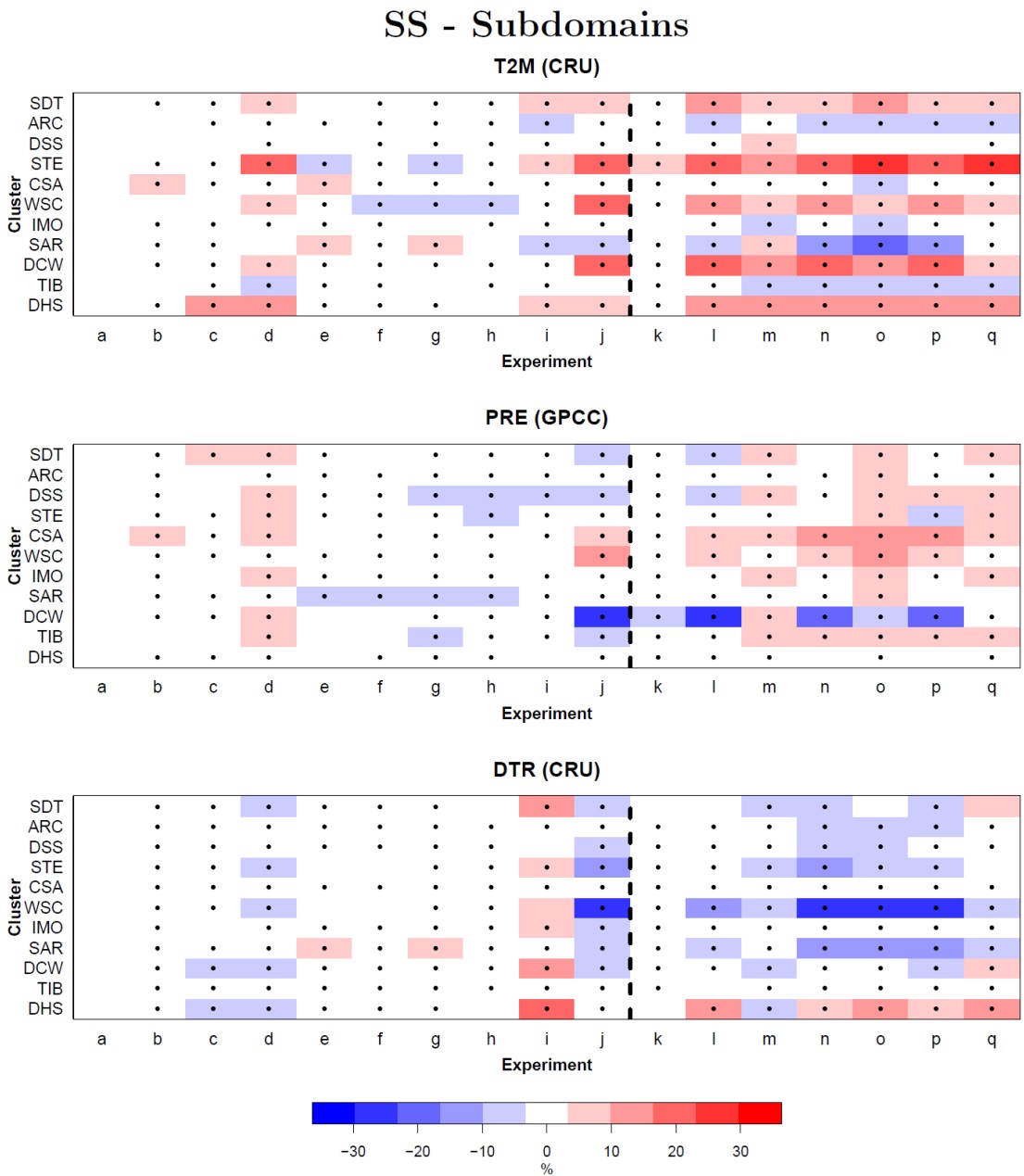

**Figure 8.** Skill Score (SS) derived from the MAE calculated for each domain sub-region over the monthly climatological values of the seasonal cycle of different COSMO-CLM simulations and observational datasets. From top to bottom, each panel represents the results obtained for near surface temperature (T2M) using the CRU, for precipitation (PRE) using the GPCC and for diurnal temperature range (DTR) using the CRU as observational datasets. Positive (negative) values indicate better (worse) performances of the considered simulations compared to the reference one. The points in correspondence of different clusters and experiments indicate that the change is the SS is the same in sign, among the three observational datasets considered for each variable.

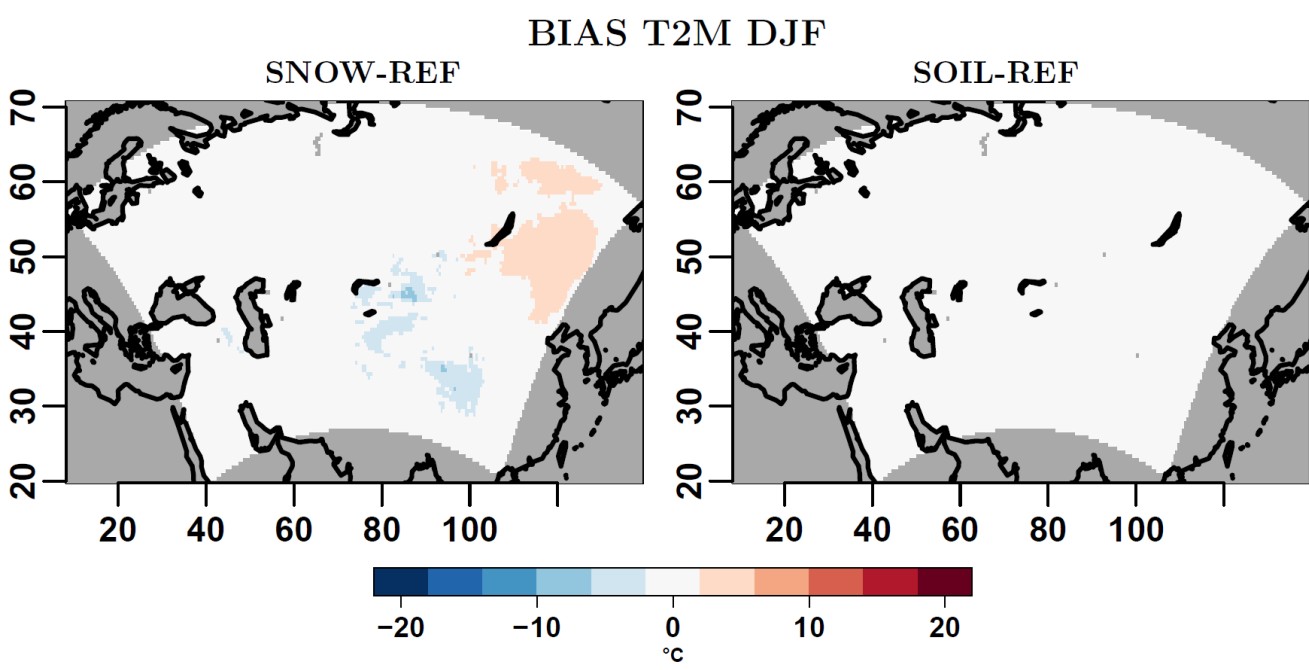

**Figure 9.** Near surface temperature (T2M, °C) winter mean bias calculated with respect to the reference simulation over the period 2006-2015, for the simulations **SNOW** (*left*) and **SOIL** (*right*).

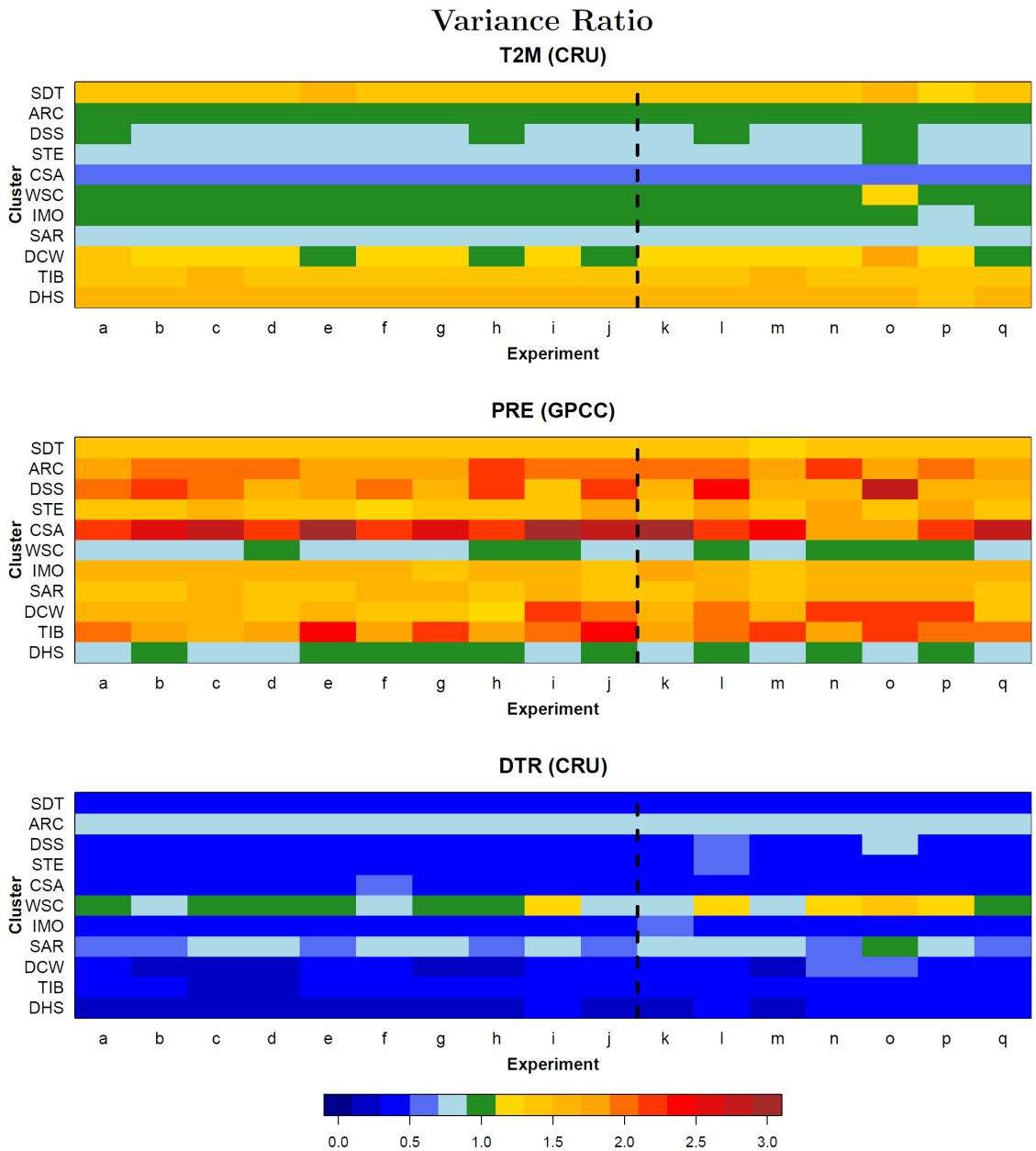

**Figure 10.** Fraction of variance calculated between the monthly anomalies over the period 1996-2005 derived from the different COSMO-CLM simulations and a single observational dataset, for (top to bottom) for near surface temperature (T2M) using the CRU, for precipitation (PRE) using the GPCC and for diurnal temperature range (DTR) using the CRU as observational datasets, for each of the 11 sub-regions obtained by k-means clustering. The dotted vertical black line divides the simulations with the same configuration of the reference simulation plus a single change in the model setup (*left*) and the ones obtained through the combinations of the previous ones (*right*). Values larger (smaller) than one indicate better (worse) performances of the considered simulations with respect to the reference one, in the representation of observed variability.

# Amplitude Variance Ratio Changes OBS vs SIM

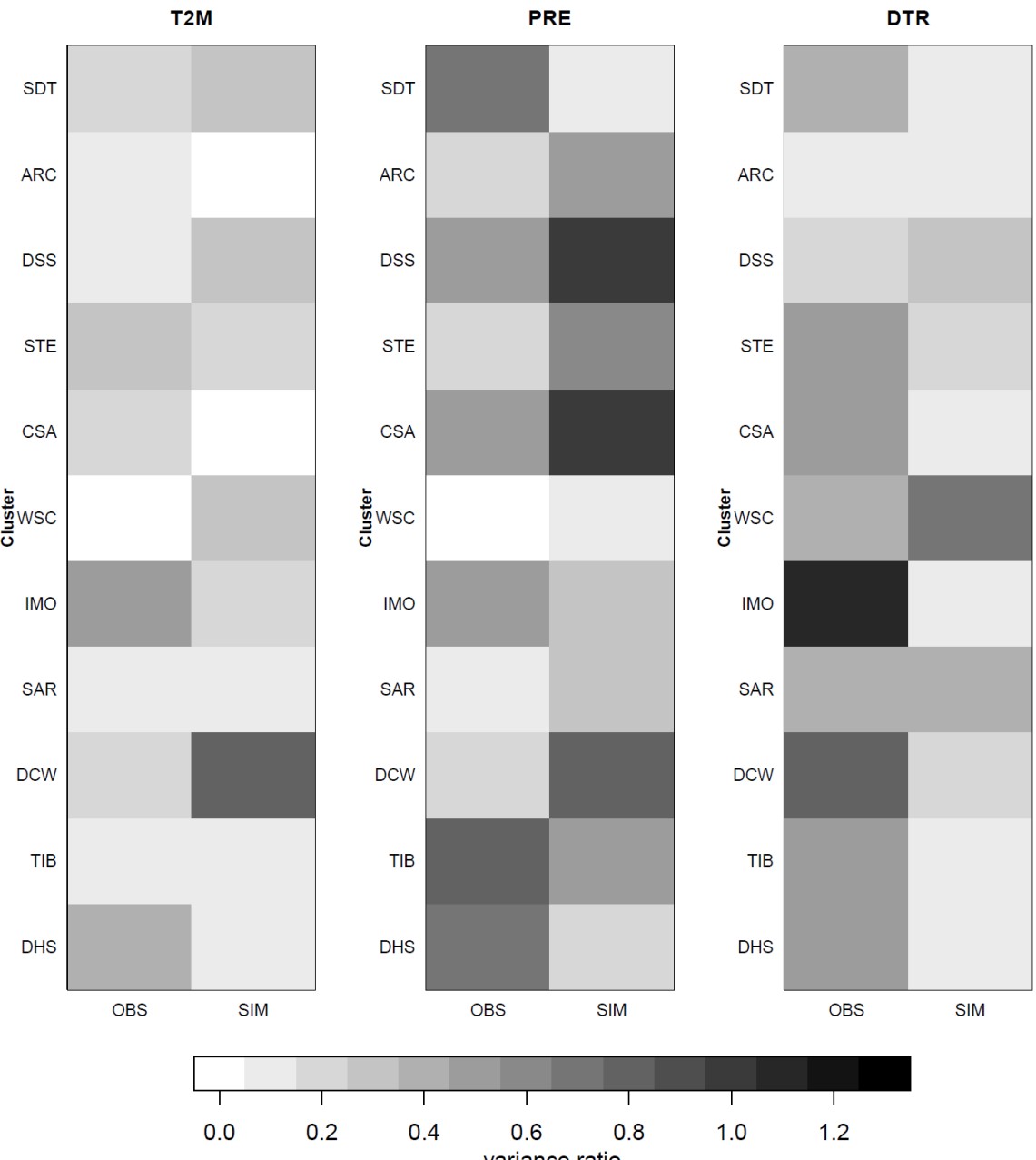

**Figure 11.** Absolute differences in variance ratio. The left column of each panel shows the absolute differences in the variance ratio of the same experiment **a** when considering different observational datasets. The right column of each panel shows the range of absolute differences betweeen the variance ratio of experiment **a** and the one of the other experiments, from **b** to **q**, when using a single observational dataset for each variable. Each of the three panels, from left to right, presents, respectively, the results of the comparison for near surface temperature (T2M), precipitation (PRE) and diurnal temperature range (DTR). The different rows show the results obtained for each of the clusters.

**Table 1.** List of performed experiments and their corresponding configuration.

| Experiment | Changes in Model Configuration |
| --- | --- |
| **a** | Reference Simulation - CORDEX East Asia setup |
| **b** | **a**+AEROCOM Aerosol Dataset (Kinne et al., 2005) |
| **c** | **a**+Surface albedo determined as a weighted mean of two external fields for dry and saturated soil |
| **d** | **a**+Vegetation albedo modified considering forest fraction. |
| **e** | **a**+Type surface-atmosphere transfer based on prognostic TKE in the surface layer |
| **f** | **a**+Cloud representation taking into account subgrid-scale condensation; cloud cover and water content calculated according to a statistical closure. |
| **g** | **a**+Equal to **f** but cloud cover and water content calculated according to a relative-humidity criterion |
| **h** | **a**+Exponential root distribution |
| **i** | **a**+Soil heat conductivity taking into account soil moisture/soil ice |
| **j** | **a**+Hydraulic lower boundary considering ground water with drainage and diffusion |
| **k** | **a+e+f+g** |
| **l** | **a+h+i+j** |
| **m** | **a+b+d** |
| **n** | **a+d+h+i+j** |
| **o** | **a+b+d+e+f+g+i** |
| **p** | **a+b+d+e+f+g+h+i+j** |
| **q** | **a+d+i** |
|  |  |
| **a_ERAInterim** | **a** driven by ERAInterim |
| **q_ERAInterim** | **q** driven by ERAInterim |
|  |  |
| **TIMESTEP** | **a**+120s Timestep |
|  |  |
| **SOIL** | **a**+increased soil layers number and depth (**25-year long**) |
| **SNOW** | **a**+increased soil layers number and depth + use of multi-layer snow model (**25-year long**) |
|  |  |
| **a2** | **a**+initial date shifted by +1 month |
| **a3** | **a**+initial date shifted by +3 month |
| **a4** | **a**+initial date shifted by -1 month |
| **a5** | **a**+initial date shifted by -3 month |

**Table 2.** General description of model setup of the reference simulation **a**.

| | |
|---|---|
| **Spatial Resolution** | $\sim$25 km |
| **Timestep** | 150s |
| **Domain Extent** | 326$\times$220 points |
| **Convection** | Tiedke |
| **Time Integration** | Runge-Kutta, |
| **Lateral Relaxation Layer** | 250km |
| **Soil Model** | TERRA-ML SVAT |
| **Aerosol** | Tegen (Tegen et al., 1997) |
| **Albedo** | Function of soil type |
| **Rayleigh Damping Layer (rdheight)** | $\geq$ 18km |
| **Active Soil Layers** | 9 |
| **Active Soil Depth** | 5.74m |
| **Atmospheric Vertical Layers** | 45 |

**Table 3.** Sub-regions resulting from the k-means clustering based on climatological monthly means of near surface temperature (T2M, CRU), precipitation (PRE, GPCC) and diurnal temperature range (DTR, CRU) for CORDEX Central Asia. Acronyms are assigned to the different regions corresponding to their main climatic characteristics. Together with the names, mean climatic information is provided for each area. The regions illustrate the wide range of climate zones of the Central Asia domain.

| Region | max T2M | min T2M | mean T2M | max PRE | min PRE | mean PRE | max DTR | min DTR | mean DTR |
|--------|---------|---------|----------|---------|---------|----------|---------|---------|----------|
| SDT | 22.7 | 0.5 | 11.8 | 49.1 | 10.7 | 29.5 | 13.9 | 8.5 | 11.4 |
| ARC | 12.5 | -22.6 | -5.5 | 61.0 | 22.6 | 36.0 | 8.9 | 4.5 | 7.0 |
| DSS | 18.4 | -21.3 | -0.3 | 76.2 | 4.7 | 26.4 | 14.5 | 11.1 | 13.0 |
| STE | 22.5 | -10.1 | 6.7 | 31.1 | 15.2 | 22.0 | 13.8 | 8.5 | 11.4 |
| CSA | 16.8 | -33.1 | -8.0 | 55.0 | 10.1 | 28.0 | 15.4 | 8.2 | 11.3 |
| WSC | 19.3 | -5.2 | 6.7 | 65.7 | 30.7 | 44.4 | 10.8 | 5.3 | 8.3 |
| IMO | 22.3 | 3.3 | 13.8 | 152.0 | 9.8 | 62.3 | 11.7 | 9.0 | 10.5 |
| SAR | 17.8 | -17.6 | 0.0 | 71.9 | 26.3 | 45.3 | 11.3 | 6.3 | 9.1 |
| DCW | 21.5 | -11.5 | 6.1 | 32.9 | 2.5 | 12.4 | 14.5 | 12.3 | 13.7 |
| TIB | 9.4 | -12.5 | -1.2 | 94.4 | 3.0 | 35.3 | 16.4 | 11.8 | 14.1 |
| DHS | 29.4 | 9.0 | 19.9 | 27.5 | 3.4 | 12.7 | 14.9 | 11.1 | 13.3 |

**Table 4.** Changes of the SS calculated for the experiment **q** with respect to reference simulation **a**, when both simulations are driven by ERA-Interim (*left*) and NCEP2 (*right*) reanalysis data.

| Obs. Dataset | T2M | | PRE | | DTR | |
|:---:|:---:|:---:|:---:|:---:|:---:|:---:|
| | ERAI | NCEP2 | ERAI | NCEP2 | ERAI | NCEP2 |
| CRU | +7.5 | +4.9 | +3.5 | +3.4 | +1.8 | +1.1 |
| UDEL | +7.3 | +5.7 | +3.8 | +3.5 | x | x |
| GPCC | x | x | +4.1 | +3.6 | x | x |
| MERRA2 | +11.1 | +7.4 | x | x | +3.1 | +2.4 |
| ERAINT | x | x | x | x | +1.5 | +1.4 |