# Peer review of "Sensitivity studies with the Regional Climate Model COSMO-CLM 5.0 over the CORDEX Central Asia Domain"

_Geoscientific Model Development, 2019_

## Short Comment (SC1) · 29 Apr 2019

This manuscript currently has a very short data availability section which simply states that the data is available on request. Under the current GMD editorial policy (https://doi.org/10.5194/gmd-8-3487-2015), authors are expected to publish the data on which their manuscript depends, unless this is not possible for reasons such as third party copyright constraints.

In this case, COSMO is a proprietary model so it is not possible to publish the model source. However the authors should explain that COSMO is free for academic use, and point to the site to obtain a licence.

For a model evaluation paper, the configuration files used to drive the model are also critical and should be publicly archived and referred to if this is legally possible. In addition, it is apparent that a lot of data processing was required in order to obtain the model inputs and analyse the outputs. All of this code should be publicly archived if legally possible so that readers can reproduce the work here, and discover the reasons for any surprising results.

The expected standard is that the code and data section will point to public archives for everything a reader would need to reproduce the results in the paper. Where this is not possible (for example because of third party licence constraints), the reasons should be given explicitly, and every effort made to correctly identify the exact version of all the code and data so that a reader who can obtain the required licences is able to access exactly the right code.

———————————————

---

## Referee Comment (RC1) · Anonymous Referee #1 · 2 May 2019

The manuscript presents results of a sensitivity experiment for setting up the regional climate model - COSMO-CLM in the CORDEX Central Asia domain. The experiment allows finding a combination of different physical parameterizations, which gives the best CCLM performance over the considered domain. Such sensitivity experiments are the first and necessary step before starting downscaling of multi-decadal GCM climate projections and aim to understand what regional processes are more important in different CORDEX domains. A lot of work has been done and the results of this study are certainly interesting to the CORDEX community.

Although these general positive considerations, I regret to inform you that I cannot recommend the publication of the manuscript in its present form, being the manuscript affected by serious issues that needs careful attention, as highlighted in the General comments.

**General comments**

The level of the manuscript is quite poor; especially the Introduction is too "educational" and seems more similar to a technical tutorial for PhD students than to a scientific paper for experts. Many concepts are explained in details, but they are well known by the scientific community working with RCMs, and could be replaced by proper references. On the other side, a detailed synthesis of the state of art is completely missing, especially for what concerns previous sensitivity studies performed with CCLM or other regional models. There are specific works by Bellprat et al. (2011, 2012) or Bucchignani et al. (2016) that have been referenced, but were not mentioned properly. In the COSMO consortium, specific Priority Projects (CALMO, CALMO-MAX) have been established in order to optimize the model configurations, but they were not mentioned. In this view, I suggest the following papers to be analyzed and referenced:

Avgoustoglou E, Voudouri A, Khain P, Grazzini F, Bettems JM (2017) Design and Evaluation of Sensitivity Tests of COSMO model over the Mediterranean area. Perspectives on Atmospheric Sciences, 1:49-55

Voudouri A, Khain P, Carmona I, Avgoustoglou E, Kaufmann P, Grazzini F, Bettems JM (2018) Optimization of high resolution COSMO model performance over Switzerland and Northern Italy, Atmospheric Research, 213:70-85.

My major concern is however related with robustness and significance of the results. A key aspect is that none of the simulations represents correctly the observed fields used as reference. Consequently, there is no value in analyzing the relative performance of the simulations, simply because all do in a terrible bad way. A temperature bias larger than 15° C or a precipitation bias larger than 200% is not acceptable.

A key question is about the reason of this shortcoming. Is it due to errors in the reference CCLM model configuration? A key parameter is certainly the time step adopted (dt), whose value is not specified in the manuscript, but it needs particular care. Alternatively, is it due to errors in the boundary conditions? The authors decided to use NCEP2, but I personally would prefer ERA-Interim, which are characterized by higher resolution, so reducing the resolution jump (other critical aspect).

The paper does not investigate the origins of these strong biases. Section 3 contains only a rough (boring, in some places) description of the figures, but no significant insights are provided. Some considerations are provided in the Conclusions, but this is not the right place, since Conclusions should be focused on the benefit of sensitivity runs with respect to the reference one.

In Sec. 2.3, you have properly defined some subdomains, but then they are used only for the analysis of variance. Instead, the results in terms of MAE (presented in Figs. 6 and 7) are averaged over the whole domain, which is too big and includes very different climate areas. I recommend that further investigations in terms of MAE be performed considering the single subdomains.

Finally, the differences among observational datasets are not discussed and the possible reasons for these discrepancies are not investigated.

**Specific comments**

Pag 2, Lines 11-18: "Among the… at once". This paragraph contains too many geographical and economical details and in my opinion does not fit well into the Introduction.

Pag 2, Lines 25–33 and Pag 2 Lines 1-9: "The countries… due to climate warming". These paragraphs are rather an analysis of the implications of climate changes on this area, and in my opinion do not fit well with the aim of the work. They should be significantly shortened.

Pag 2, lines 9-11: "All the reported… strategies". This sentence is a repetition of concepts already expressed.

Pag 2, line 14 "Assessing …. evaluation". This definition is well known by climatologists and can be removed.

Pag 2, line 17: "Model evaluation… development". This sentence is prosaic and can be removed.

Pag 3, line 27: "A series… simulation". This sentence is prosaic and can be removed.

Pag 4, lines 8-11: "Such analyses … configurations". This concept has already been explained previously and can be removed.

Pag 4, lines 11-14: "In the light… are presented". From this sentence, I do not see a relationship between your sensitivity and the CORDEX-CORE activities. Please explain better this relation and, at the same time, explain what CORDEX-CORE is.

Pag 4, line 32: "This study… domain". This concept has already been expressed in the Introduction. Please put it only once.

Figure 1: It is preferable to show the domain using the geographical coordinates, since the reader is generally not interested in the rotated coordinates (being rotated coordinates used only internally for COSMO-CLM calculations).

Pag 5, lines 11-13: "The model configuration… .en).". These details are not necessary, especially because readers are generally not authorized to download the model configuration from the website of CLM Community. Please add more details about model configuration in Table 2.

Table 2 (caption): The general description of model setup of the reference simulation is very poor. It contains details that have already been explained in the text (e.g. spatial resolution, domain extent). Btw, the domain extent must be expressed in terms of max/min longitude/latitude and not in terms of number of grid points. Some important details of the model setup are missing. For example, in Table 1 you write that in **b** configuration Tegen aerosol is used , but what is the aerosol

used in **a**? I guess the default Tanre, but you have to specify it. Similarly, for albedo: what is the default one? I guess albedo as function of soil type.

Pag 5, lines 19-20: "since their... simulations". This is not a good reason to employ NCEP reanalysis as driving data. Generally, data at higher resolution are preferred. Btw, what is the resolution of GCM normally employed in CORDEX simulations?

Sec 2.2: It is not clear if the original resolution of datasets CRU, UDEL, MERRA GPCC is 0.5° or if they have been interpolated on a common grid with 0.5° resolution.

Pag 6, lines 19-20: "The Climate ... interpolation". This technical detail (usage of CDO) is not necessary and can be removed.

Pag 7, line 25: It is not specified if variances (observed and simulated) are evaluated starting from monthly values.

Pag 8, line 12: A bias of 15° or larger is not acceptable.

Pag 8, line 12-21: In this paragraph you are commenting Figure 3, which is related to simulation **a**, so it is not wise to comment here also the simulations SOIL and SNOW.

Pag 9, line 1: Why do you claim that this sentence is "interestingly" ?

Pag 10, line 10: Why in this case analyses focused on a single observational dataset?

Pag 11, lines 17-18: "For the experiments... experiment **q**". This sentence is just a repetition of the sentence at lines 13-15. Please combine them.

Pag 11, line 24-25: "This indicates...driving dataset". This sentence is very strong and must be supported by results that are more robust. The few numbers shown in Table 4 are not sufficient. Moreover, you should add in Table 4 the improvements achieved when using NECP2, in order to have a quantitative comparison.

Pag 11, lines 32-33: "Values closer... observations". These statements are obvious and can be removed.

Minor corrections

Pag 6, line 23: You have already explained that the reference configuration is the **a**. Please remove "(a, Tab.2)".

Pag 7, line 6: Do you mean Tab. 3 (instead of Tab .4)? Otherwise, Tab.3 is never referenced.

Title of Fig.6: If SS is defined according with equation (1), why did you add (%) next to SS?

Pag 11, lines 16-17: In the title you use NCEP, in the text NCEP2, please use always the same acronym.

---

## Referee Comment (RC2) · Anonymous Referee #2 · 1 Jun 2019

Review of "Sensitivity studies with the Regional Climate Model COSMO-CLM 5.0 over the CORDEX Central Asia Domain" by Russo et al.

General Comments:
In order to provide reliable future climate projections, the model should be able to capture the present climate feature realistically. For seeking the optimal setups for regional climate model COSMO-CLM over the CORDEX Central Asia domain, the authors have conducted series of sensitivity simulations for historical periods. With different observation/reanalysis dataset as references, they evaluated the general model performance in capturing the mean climate and variability of temperature, precipitation and daily temperature range and figured out the relative optimal model setups for CORDEX Central Asia domain.

Though the study is rather regional specific, it is believed to be interesting for the regional climate modelling community. The manuscript is in general well organized. The methods used are reliable and language is good. However, the manuscript suffers from some major problems. The authors will need to address them before the manuscript can be considered for publication in Geoscientific Model Development.

Specific Comments:
(1) It is suggested to reduce to a relative brief introduction about vulnerability of CORDEX Central Asia to the effects of climate changes, say from Page 2 Line 19 to P3 L11. Furthermore, there is a general lack of reviewing studies about model performance evaluation, which are related to the experimental setups, assessment methods and discussion, c.f., Li et al. (2018) and Huang et al. (2015) and so on:
Li, D., Yin, B., Feng, J., Dosio, A., Geyer, B., Qi, J., ... & Xu, Z. (2018). Present Climate Evaluation and Added Value Analysis of Dynamically Downscaled Simulations of CORDEX—East Asia. Journal of Applied Meteorology and Climatology, 57(10), 2317-2341.
Huang, B., Polanski, S., & Cubasch, U. (2015). Assessment of precipitation climatology in an ensemble of CORDEX-East Asia regional climate simulations. Climate Research, 64(2), 141-158.

(2) The authors conducted a series of experiments considering different configurations, which are supposed to be significant for skills of modelling. However, some specific setups, which have been proved to be important in regional climate modelling, have not been considered in the study, such as the technique of spectral nudging (von Storch et al. 2000) and topography. RCM simulation with spectral nudging can add value in reproducing snow water equivalents, coastal winds and some meso-scale phenomena (von Storch et al. 2016), as well as annual mean temperature and precipitation (Tang et al. 2017). The reviewer suggest the authors add one experiment with spectral nudging.
In addition, about two additional 25-year long simulations covering 1991-2015, why do not use a period backward, say 1981-2005, so that there are longer spinup

time, and same comparison period as other experiments?

(3) There are some problems in Figure plottings: a). Figure 1, please plot in lon and lat dimensions rather than in rlon, rlat dimensions; b). Figure 2, it is better to add names of subregions on map rather than using a colorbar; c). Figure 3, the colorbar scheme is rather poor. It is hard to distinguish them on the map. Less and distinguishable colors are suggested to use, with more equal divisions within -10 to 10 and less divisions from (±) 10 to (±) 20.

(4) Some descriptions does not reflect the figures or tables. Such as P10 L26, I would not say experiment q in Fig.7 (upper panel) fits to the description; P10 L34, experiment o does not share the use of the setup of j. A thorough revision is needed to catch all these inconsistencies.

(5) I would not agree the conclusion that "The results for the mean climate appear to be independent of the observational dataset used for evaluation and of the boundary data employed to force the simulations". In fact, according to Fig. 3 and Fig. 5, it is clear that skill of simulated mean climate depends on the referred observational dataset. Furthermore, Li et al. JAMC (2018) clearly shows that both observational dataset and boundary forcing have impacts on the skill assessment of simulated mean climate.

(6) A temperature bias exceeding 15ºC is too large, especially when the simulations are driven by high-quality reanalysis dataset rather than by GCMs generally used by CORDEX community. A fully discussion on this issue is necessary.

(7) Only whole-region or subregion averaged values for SS or variance ration (Fig. 6 – Fig. 8) are not enough. Spatial distribution patterns of these scores are significant for a thorough model quality assessment. I would not suggest to plot every spatial distribution of these scores for each reference dataset, but representative figures are necessary, if not in the manuscript but in the supplementary part.

Minor Comments:
(1) P6 L8-15: It's better to summarize the data information in a table.
(2) P7 L6: Tab. 3 not Tab.4, the same for P9 L6 and P12 L14
(3) P7 L7-8 Combine two paragraphs into one
(4) P7 L13: 'Mean Absolute Error' to 'Mean Absolute Error (MAE)'
(5) P11 L24-25: It may be only appropriate when you run CCLM driven by similar high quality reanalysis datasets.
(6) P12 L3-19: Please indicate which subpanel of Figure 8 you are describing in the text.
(7) P12 L26-27: range of absolute differences instead of absolute differences?

---

## Author Comment (AC1) · 18 Jul 2019

Dear David Ham,

thank you very much for your comment to our manuscript.

A similar concern about availability of data was already raised by the manuscript editor.

We think that availability of all the configuration files and postprocessing routines that you mentioned is really important for reproducibility purposes. For this reason, we will be more than happy to make publicly available on Zenodo all these data at the moment of the submission of the final version of the paper.

At the same time, we will provide information about COSMO-CLM and on the fact that it is free for academic purposes, in the final version of the manuscript.

Thank you again for your important comment.

On behalf of the all authors, Emmanuele Russo
* * *

---

## Author Comment (AC2) · 18 Jul 2019

[11pt]article epsfig color url gensymb textcomp

Reply to
**1st Reviewer**
*Russo, E., Kirchner, I., Pfahl, S., Schaap, M., and Cubasch, U.: Sensitivity studies with the Regional Climate Model COSMO-CLM 5.0 over the CORDEX Central Asia Domain, Geosci. Model Dev. Discuss., https://doi.org/10.5194/gmd-2019-22.*

[Figure]

Dear reviewer,

Thank you very much for your effort in reviewing our paper.

Below we go point by point through your technical corrections, presented in *italic*, detailing how we dealt with your concerns reported in *Bold*.

Please, find an additional version of the response attached as supplement material, where the caption of the figures is reported in its integer form.

Thank you.

*General Comments*

- *T*he level of the manuscript is quite poor; especially the Introduction is too "educational" and seems more similar to a technical tutorial for PhD students than to a scientific paper for experts. Many concepts are explained in details, but they are well known by the scientific community working with RCMs, and could be replaced by proper references. On the other side, a detailed synthesis of the state of the art is completely missing, especially for what concerns previous sensitivity studies performed with CCLM or other regional models. There are specific works by Bellprat et al. (2011, 2012) or Bucchignani et al. (2016) that have been referenced, but were not mentioned. In this view, I suggest the following papers to be analyzed and referenced:

  Avgoustoglou E., Voudouri A., Khain P., Grazzini F., Bettems J.M.: Design and

Evaluation of Sensitivity Tests of COSMO model over the Mediterranean area. Perspectives on Atmospheric Sciences, 1:49-55.

Voudouri A., Khain P., Carmona I., Avgoustoglou E., Kaufmann P., Grazzini F., Bettems J.M. (2018): Optimization of high resolution COSMO model performance over Switzerland and Northern Italy, Atmospheric Research, 213:70-85.

**We agree with the referee and think that the introduction could be improved and made more easily readable. Also following the suggestion of the second referee, sensibly shortening the part of the introduction on the description of vulnerability of Central Asia to the effects of climate change should help in this sense. At the same time, we will try to expand the part of the introduction on the state of the art of model performance evaluation and model calibration. For this we will consider the suggested references, together with additional ones. We will additionally try to express concepts well known in the community in a shorter form, when possible.**

- *M*y major concern is however related with robustness and significance of the results. A key aspect is that none of the simulations represents correctly the observed fields used as reference. Consequently, there is no value in analyzing the relative performance of the simulations, simply because all do in a terrible bad way. A temperature bias larger than 15 °C or a precipitation bias larger than 200 % is not acceptable.

  A key question is about the reason of this shortcoming. Is it due to errors in the reference CCLM model configuration? a key parameter is certainly the time step adopted (dt), whose value is not specified in the manuscript, but it needs particular care. Alternatively, is it due to errors in the boundary conditions? The authors decided to use NCEP2, but I personally would prefer ERA-Interim, which

are characterized by higher resolution, so reducing the resolution jump (other critical aspect).

**The concerns raised by the referee are very important. Despite we think that we partly tried to take care of them already in the manuscript, at least partially in the experiments design, we recognize that we were not very careful in the treatment of some points, not clearly specifying them in the text. Trying to answer to the referee comment, in a first place we want to highlight the fact that the paper, given the large amount of tests conducted, is very important in order to understand whether the evinced biases are characteristic of the model itself and if they can be reduced by properly configuring the model and to which degree. For this, in our experiments we tried to be careful in as many points as possible as to isolate different sources of uncertainties. For this purpose, we performed additional simulations driven by ERA-Interim to test the effect of different boundaries. The results of these simulations are not particularly different than the ones driven by NCEP2, presenting a very similar bias with the considered observations, for all variables (Fig. ??). We realized that we did not carefully discussed this point in the paper and we will better highlight it in the new version of the manuscript. With this additional simulations we aimed to show that the given biases are not due to the effect of considered boundary data. Or at least that, in our case, the use of mainly employed higher resolution reanalyses such as ERAInterim as boundaries, does not affect significantly model performances. Additionally, the improvement of the results when using NCEP2 or ERAInterim, with the finally proposed configuration (experiment q), is also very similar in the two cases (Table 4 of the original manuscript). We want to claim the fact that we decided, very consciously, to use NCEP2 instead of normally employed ERAInterim, for a clear reason: to try to reproduce the resolution jump (mentioned by the referee) that we will have using GCMs for the CORDEX-CORE simulations. In fact, the plan within the COSMO-CLM community is to use, for the CORDEX-CORE simula-**

tions, 3 GCMs: MPI-ESM, HadGEM and NorESM. Their resolution is respectively around 210km, 210x140km and 270x210 km. In this sense, also considering the results of the mentioned simulations, we think that our choice of using NCEP2 instead of ERAInterim is more than justifiable. We realized that we were not accurate enough on this point in the text. In the final version of the manuscript we will include information on the three models and (as supplementary material) the picture of the bias of the ERA-Interim driven simulation, together with related information. Other conducted simulations that we performed and that could help in isolate the reasons for the model bias are the two additional simulations that we performed with diffetrent soil layers and the CCLM multi-layer snow model (Fig. ??). These results are important because they show that it is not possible to improve model performance in terms of winter temperature over Siberia following previously suggested hypotheses (the snow model produces even warmer biases over the area in winter), and the given bias is very likely due to the model formulation itself. This is very important because it highlights the necessity to put more efforts in the improvement of the simulation of snow processes and permafrost in COSMO-CLM. We will provide the new figures of the bias of winter temperatures for these simulations in the supplementary part of the final version of the paper. The only model uncertainty factor that we did not consider in our former version of the manuscript is the different time step. For tackling this issue, following the referee comment, we now conducted a new simulation with a different timestep of 120s instead of 150s. The biases against observations calculated for this new simulation are reported in Fig. ?? (in this case only CRU is used for T2M and DTR, while GPCC is considered for PRE). As you can see, using a different time-step the results do not change significantly. This confirms even more that the given biases are characteristic of the model and do not depend upon the referee suggested sources of uncertainty. We will add this figure as supplementary material to the new version of the paper, and will better discuss it in the final manuscript.

Beside these considerations, the most important thing that must be considered in order to properly address the referee comment concerning the validity and significance of the results is the comparison and proper discussion of the different observational datasets. It is certain that the evinced model biases are quite remarkable in some cases, in particular during winter for temperature. For precipitation, the biases exceed 100% over large parts of the domain, but not 200% as suggested by the referee. This is normal for many RCM studies, especially for the Tibetan Plateau, as already indicated in the former version of the manuscript. Biases of 100% in precipitation are evident in most of the CORDEX simulations, for different domains and models (Kotlarski et al. 2014, Russo & Cubasch 2016, Park2013, Martynov2013). In any case, these biases, supported by the already mentioned analyses, certainly confirm the importance of trying to investigate different model configurations in order to improve model performance over the area and to determine to which degree this is possible. We think that the paper definitely gives a significant contribute in this sense. On top of that, we agree with both the referees that the different observational datasets need more attention in the manuscript and their main drawbacks and differences need to be properly discussed. For this purpose we propose to include in the final version of the manuscript Fig. ?? of the current response, showing the spread of the different observational datasets, for each variable. This will contribute to support the validity of the presented results. In fact, the spread of the different observations is larger in correspondence of those points characterized by particularly complex topography, for which model biases seem to be more remarkable, exceeding 15 °C in the case of temperature. This suggests that the particularly high biases evident in the model are hard to be quantified for these points. Additionally, if we now consider the new figures of temperature bias (Fig.??, together with the corresponding Fig. ?? for the DTR bias), that we drew considering the suggestion of the second referee to use a colorbar with fewer breaks, we can see that the very high temperature biases exceeding in some cases 15 °C are

mainly relative to the UDEL dataset and, in general, particularly large biases are limited to a few points characterized by particularly high topography, where the gridded datasets are less reliable. If we consider the CRU dataset, that is one of the most employed dataset for the area, for evaluating the results of RCMs, we see that the values of the bias rarely exceed (are below) 10 (-10) ℃, for really few points. Beside these points, still some remarkable biases are present but we think that these are well within the ranges of model simulations produced in CORDEX. For example, for the East Asian CORDEX domain the studies of Wang et al. 2013 and Bucchignani et al. 2014 showed that the CCLM over the Eastern Part of the Tibetian Plateau has a cold bias in winter, much lower than -6 C. The other area for which largest biases are present for temperature in our simulation is Western Siberia. For this region, one of the only references is the one of Ozturk et al. 2015. Their results are part of already published CORDEX results. From these, using REGCM4 with a resolution of ˜50 Km, they obtained a model climatological bias for winter temperature exceeding 8 ℃ over the area, when using CRU as a benchmark for the evaluation of their results. This is very similar to our case. Following this discussion, we propose to introduce in the new version of the paper the figure with the spread of observational datasets, together with an appropriate discussion on the possible reasons for their differences, and the new plots for the map of the bias of the reference simulation. One important thing that also needs to be highlighted in the text is certainly the fact that, beside high biases between model and observations are evident for some points, mainly where the observations are less reliable, the pattern of these biases is in general very similar among different observational datasets and well within the range of other CORDEX results. We think that this information is certainly required in order to support the significance and robustness of the results and we will include it in the new version of the manuscript.

Always in the context of significance of the results, for the rest of the analyses of climatological values, we proceeded separately considering the differences among different observational datasets. In fact, we calculated for each dataset independently the MAE and the relative skill score with respect to the reference simulation, with the aim of choosing the best performing configuration that shows the same positive effect among all the different observational datasets. We will try to express this more carefully in the paper, where we think it was not carefully stated before. Additionally, we propose to introduce Fig. ?? as a supplement to the final version of the paper. This is obtained in a similar way as in Bellprat et al. 2012, but only considering the climatological mean of each month over the considered simulation period. Basically, we selected a reference dataset for each variable (CRU for T 2M, GPCC for PRE and CRU for DTR) and then we calculated the MAE in each case, weighting the absolute error over each point by the sum of two uncertainty sources, one given by the standard deviation of the observational datasets and the other by a standard deviation representative of considered model errors, calculated among the climatological means of the reference simulation, the one with a different time step and the reference simulation driven by ERAInterim. This is the formula we used for the new plot:

$$MAE_w = \frac{1}{TJI} \sum_t^T \sum_j^J \sum_i^I \frac{|m_{t,j,i} - o_{t,j,i}|}{1 + \sigma_{o_{t,j,i}} + \sigma_{iv_{t,j,i}}} \tag{1}$$

where $t$ represents the considered month and $j$ and $i$ are the spatial indices of the points of the domain. The two terms *m* and *o* are, respectively, the model and the observational monthly means calculated for each month of the domain. $\sigma_o$ represents the mentioned observational uncertainty and $\sigma_o$ the one of the model. The 1 in the denominator has been added in order to avoid infinitive numbers when the uncertainties are considerably close to 0. In this way, points of the domain

with higher uncertainties will receive a lower weight in the computation of the MAE. As it is possible to see from Fig. ?? of this document, the results of the SS using the weighted MAE are approximately the same as in the original figure 6 of the manuscript. Some differences are present in some isolated cases, mainly for precipitation, due to high uncertainties over some area. Nevertheless, the new analyses overall confirm the results of the plot of the total skill score of the paper: simulation q has the best performance for the domain, for all variables, and most of all coherently among different observational datasets and also considering different model uncertainties. All the analyses, considering different uncertainty sources in a different way, give the same results.

In conclusion, in the light of the presented discussion, better considering the mentioned points and different sources of uncertainty, we can affirm that our results are very important and most of all significant for the improvement of model performance over the area. Generally, the model, excepted some isolated points for which the bias against observations is not really quantifiable due to the large spread of the observations, does not perform particularly worse than other models normally considered in CORDEX. For sure there is some limited number of points for which climatological biases are very high, but these seem to be related to observation uncertainty. An appropriate discussion on the sources of these biases, most likely related to the complex topography of some region, will be included in the final version of the paper.

- *T*he paper does not investigate the origins of these strong biases. Section 3 contains only a rough (boring, in some places) description of the figures, but no significant insights are provided. Some considerations are provided in the Conclusions, but this is not the right place, since conclusions should be focused on the benefit of sensitivity runs with respect to the reference one.

**We agree with the referee that the results part is in some cases boring and we will revise the text to make it more easily readable. For this purpose, we propose to place the figure on the investigation of seasonal biases to the supplement part, replacing it with the referee proposed analysis of regional behavior of the different simulations. We think that in this way some not particularly relevant part of the text could be summarized in a simpler way, while focusing on single regions could help supporting the given conclusions on best model configuration. Concerning the current structure of the paper, we actually want to keep the results part as a description of the figures (the results) as we conceived it in a first place. On the other hand, we want to keep the part on the discussion of the results in the conclusion, but, following the referee comment, changing the title into a more appropriate "Discussion and Conclusion" section.**

- *I*n Sec. 2.3, you have properly defined some subdomains, but then they are used only for the analysis of variance. Instead, the results in terms of MAE (presented in Figs.6 and 7) are averaged over the whole domain, which is too big and includes very different climate areas. I reccomend that further investigations in terms of MAE be performed considering the single subdomains.

**We agree with the referee that further investigation in terms of MAE performed considering single sub-domains is required. This in fact could help to have a more proper idea of how model results change for different areas characterized by different climate conditions, contributing to the determination of an optimal model configuration and to better discriminate reasons of possible shortcomings. Therefore, we propose to introduce Fig. ?? of the current document, representing the SS of the MAE calculated over single sub-domains, in the final version of the paper, together with an appropriate discussion of the results. The proposed figure is obtained in the same way as in the case of the entire domain:**

[Figure]

the calculation of the MAE is conducted separately for the different observational datasets. For visualization reasons, we propose to plot the results of the analysis per sub-region for a single observational dataset for each variable ( CRU for T 2M and DTR and GPCC for PRE ), with a point for each region for which the given configuration produces the same model response among the different observational datasets. Fig. ?? shows the SS of the MAE calculated for different sub-regions for all the considered observational datasetes. We propose to include this figure in the supplement part of the manuscript. At the same time we propose to also include as supplement to the final version of the paper, Fig. ?? of the current document, presenting the same analyses per sub-region but using the weighted MAE. As we can see from Fig. ??, Fig. ?? and Fig. ??, beside some exceptions, the results have a similar behavior for all different cases, with the same conclusions that can be drawn. These plots help because they allow to investigate model behavior for single regions, as already said. In particular, they allow to see that a complete improvement of model results over all the sub-regions is not completely achievable. As discussed in the introduction of our paper, one has always to be aware of the fact that calibrating the model could lead to better results, but this might also be the result of compensating errors. Reinforcing these thoughts, we think that with the proposed optimal configuration q the model improves in large part of the cases, for all variables. These results highlight again the advantages of using configuration q for the region. The newly proposed analyses also allow to see that in some cases model improvements almost reach 35% with respect to the reference simulation. This, once again, adds significance to the proposed results. In the final version of the manuscript we will change the results part as already proposed, substituting the plot of seasonal results with the one of the analyses for sub regions. The text will be changed accordingly to the new introduced results, trying to make it more easily readable.

- *F*inally, the differences among observational datasets are not discussed and the possible reasons for these discrepancies are not investigated,

**We agree with the referee. As already stated above, we aim to introduce in the new version of the manuscript a proper discussion on the different observational datasets, their differences and the possible reasons for them.**

*Specific Comments*

- *P*ag 2, Lines 11-18: "Among the...at once". This paragraph contains too many geographical and economical details and in my opinion does not fit well in the Introduction.

**We understand this issue, raised by both the referees. We realize that this part should be significantly shortened, being only secondary to the objectives of the paper. We think that this would also help making the introduction more easily readable.**

- *P*ag 2, Lines 25-33 and Pag 2 Lines 1-9: "The countries...due to climate warming". These paragraphs are rather an analysis of the implications of climate changes on this area, and in my opinion do not fit well the aim of the work. They should be significantly shortened.

**We agree. We will shorten this part as proposed in the previous answer.**

- *P*ag2, lines 9-11: "All the reported...strategies". This sentence is a repetition of concepts already expressed.

**We agree. This part is repetitive and we will delete it from the final version of the
manuscript.**

- *P*ag 2, Line 14: "Assessing...Evaluation". This definition is repetitive and can be
removed

**We will remove this line accordingly to the referee comment.**

- *P*ag 2, Line 17: "Model Evaluation...development". This sentence is prosaic and
can be removed.

**We agree with the referee and will correct the text accordingly.**

- *P*ag 3, Line 27: "A series...simulation". This sentence is prosaic and can be
removed.

**Same as above.**

- *P*ag 4, Lines 11-14: "In the light...are presented". From this sentence, I do not
see a relationship between your sensitivity and the CORDEX-CORE activities.
Please explain better this relation and, at the same time, explain what CORDEX-
CORE is.

**CORDEX-CORE stands for CORDEX - Coordinated Output for Regional Evalua-
tions (CORE). This is the next phase of the CORDEX initiative, designed in the**

light of the upcoming IPCC report, with the objective of coordinating a set of high resolution climate projections for different regional domains, including Central Asia. In this perspective, our work represents the first step for the production of climate projections for the Central Asia domain using COSMO-CLM, evaluating general model performances, isolating the effects of different uncertainty sources on model results and determining an optimal model configuration for a region region for which almost no reference exists. Following the referee comment we realized that we probably did not specify very well this information in the former version of the manuscript. Consequently, we propose to extend the relative part of the text accordingly.

- *"*This study...domain". This concept has already been expressed in the Introduction. Please put it only once.

**We will remove this repetitive part of the text, following the referee comment.**

- *F*igure 1: It is preferable to show the domain using the geographical coordinates, since the reader is generally not interested in the rotated coordinates (being rotated coordinates used only internally for COSMO-CLM calculations)

**We agree. We propose a new version of the map of the domain topography, presented in Fig. ?? of the current document, in geographical coordinates. The figure caption will be modified accordingly.**

- *P*ag 5, lines 11-13: "The model configuration...en)". These details are not necessary, epsecially because readers are generally not authorized to download the

model configuration from the website of CLM Community. Please add more details about model configuration in Table 2.

**We agree with the referee that the description of the model configuration introduced in Table 2 of the previous manuscript version needs to be extended. Also, the link to the CLM-Community webpage could be removed since, as suggested by the referee, not all users could access the given configuration. Proposing to modify the text accordingly, we still want to mention the fact that we use as a benchmark for our reference simulation, the configuration of COSMO-CLM for another CORDEX domain, but covering a large part of Central Asia. This follows the main guidelines of the CLM-community for the model configuration.**

- *T*able 2 (caption): The general description of model setup of the reference simulation is very poor. It contains details that have already been explained in the text (e.g. spatial resolution, domain extent). Btw, the domain extent must be expressed in terms of max/min longitude/latitude and not in terms of number of grid points. Some important details of the model setup are missing. For example, in Table 1 you write that in b configuration Tegen aerosol is used, but what is the aerosol used in a? I guess the default Tanre, but you have to specify it. Similarly, for albedo: what is the default one? I guess albedo as function of soil type.

**We agree with the referee and will modify table 2 accordingly to his comment. Information about the time step of the reference simulation (150s), the Aerosol, for which we used TEGEN as default, and the albedo, as a function of the soil type, will be included, together with information on the domain extent expressed as min and max lon and lat.**

- *P*ag 5, lines 19-20: "since their...simulations". This is not a good reason to employ NCEP reanalysis as driving data. Generally, data at higher resolution are preferred. Btw, what is the resolution of GCM normally employed in CORDEX simulations

**As already stated above, we were completely aware of the decision taken using NCEP2 reanalyses instead of ERAInterim, with the main goal of simulating a jump in resolution more similar to the one using the GCMs mentioned above. For this, we consider our choice more than valid. NCEP2 are still considered a valuable reanalysis dataset, that has been used in a large variety of studies. Beside that, we also conducted similar analyses with ERAInterim to have an estimate of the effect of using higher resolution drivers on the results and how they change in the different cases. We demonstrated that the effect of the two different datasets on the simulation of climatological monthly means for the considered period is almost the same. We will highlight this point better in the new version of the manuscript.**

- *S*ec 2.2: it is not clear if the original resolution of datasets CRU, UDEL, MERRA GPCC is 0.5°or if they have been interpolated on a common grid with common 0.5°resolution.

**The resolution of the mentioned datasets is all 0.5°. No interpolation was needed. We will try to make it clearer in the new version of the manuscript.**

- *P*ag 6, lines 19-20: "The climate...interpolation". This technical detail (usage of CDO) is not necessary and can be removed.

**CDO is an important tool for the postprocessing of climate data, freely available. It is a personal decision, but also following the work of other papers, given its importance, we think that it deserves to be referenced in the text.**

- *P*ag 7, line 25: "It is not specified if variances (observed and simulated) are evaluated starting from monthly values.

**We acknowledge the fact that we have not been very clear in this sense in the previous version of the paper. We now propose to modify the final version of the paper better specifying that the variance is evaluated starting from monthly values.**

- *P*ag 8, line 12: A bias of 15° or larger is not acceptable.

**Again, this seems to be a problem related more to the reliability of the gridded observational datasets over some points rather than to the model itself.**

- *P*ag 8, line 12-21: In this paragraph you are commenting Figure 3, which is related to simulation a, so it is not wise to comment here also the simulations SOIL and SNOW.

We agree. We will try to introduce the results of simulation **SNOW** and **SOIL** in an additional separated subsection of the results part.

- *P*ag 9, line 1: Why do you claim that this sentence is "interestingly"?

**We think that "interestingly" in this case could be deleted.**

- *P*ag 10, line 10: Why in this case analyses focused on a single observational dataset?

In the figure of the variance ratio we showed the results just for a single observational dataset for each variable, simply for graphical reasons. Nevertheless, the same analyses were conducted for the different observational datasets, and considered when discussing the uncertainties in the estimation of simulated variance. Realizing that this was not clearly specified in the text, we will modify this part accordingly.

- *P*ag 11, lines 17-18: "For the experiments... experiment q". This sentence is just a repetition of the sentence at lines 13-15. Please combine them.

We will try to merge the two sentences together as suggested by the referee.

- *P*ag 11, line 24-25: "This indicates...driving dataset". This sentence is very strong and must be supported by results that are more robust. The few numbers shown in Table 4 are not sufficient. Moreover, you should add in Table 4 the improvements achieved when using NCEP2, in order to have a quantitative comparison.

**We agree that the given sentence is too strong. Nevertheless, it is true that the 2 conducted ERAInterim-driven simulations present similar climatological values to the NCEP2-driven ones, in both cases. Even though it is not possible to draw a general conclusion on the effects of the boundaries on COSMO-CLM for the**

**region and the given resolution, these results allow to justify the use of NCEP2 instead of commonly employed ERAInterim for the purposes of our research. We will modify the corresponding part of the text in the final version of the paper, being more careful on the conclusion we can draw from our simulations.**

- *P*ag 11, lines 32-33: "Values closer... observations". This statements are obvious and can be removed.

**We agree and will modify the text accordingly.**

*Minor Comments*

- *P*ag 6, line 23: You have already explained that the reference configuration is the a. Please remove "(a,Tab.2)".

**We agree and will modify the text accordingly.**

- *P*ag 7, line 6: Do you mean Tab.3 (instead of Tab.4)? Otherwise, Tab.3 is never referenced.

**We acknowledge the error. We referred to Tab.3. We will correct it in the final version of the manuscript.**

- *T*itle of Fig. 6: If SS is defined according with equation (1), why did you add (%) next to SS?

**Actually we propose to express SS in** %

- *l*ines 16-17: In the title you use NCEP, in the text NCEP2, please use always the same acronym.

**We will correct the error in the final version of the manuscript.**

Below we propose some additional bibliography that we will provide in the revised version of the manuscript, if not already considered, following the referee comments and the proposed discussion.

**References**

. Ozturk, T. and Altinsoy, H. and TürkeÈŹ, M. and Kurnaz, M.L., 2012. *Simulation of temperature and precipitation climatology for the Central Asia CORDEX domain using RegCM 4.0*, Climate research, 52, 63–76.

. Bucchignani, E. and Montesarchio, M. and Cattaneo, L. and Manzi, M. P. and Mercogliano, P., 2014. *Regional climate modeling over China with COSMO-CLM: Performance assessment and climate projections*, Journal of Geophysical Research: Atmospheres, 119, 121–151.

. Wang, D. and Menz, C. and Simon, T. and Simmer, C. and Ohlwein, C., 2013. *Regional dynamical downscaling with CCLM over East Asia*, Meteorology and Atmospheric Physics, 121, 39–53.

. Kotlarski, S. and Keuler, K. and Christensen, O.B. and Colette, A. and Déqué, M. and Gobiet, A. and Goergen, K. and Jacob, D. and Lüthi, D. and Van Meijgaard, E. and others, 2014. *Regional climate modeling on European scales: a joint standard evaluation of the EURO-CORDEX RCM ensemble*, Geoscientific Model Development, 7, 1297–1333.

. Park, J.H. and Oh, S.G. and Suh, M.S., 2013. *Impacts of boundary conditions on the precipitation simulation of RegCM4 in the CORDEX East Asia domain*, Journal of Geophysical Research: Atmospheres, 118, 1652–1667.

. Martynov, A. and Laprise, R. and Sushama, L. and Winger, K. and Šeparović, L. and Dugas, B., 2013. *Reanalysis-driven climate simulation over CORDEX North America domain using the Canadian Regional Climate Model, version 5: model performance evaluation*, Climate dynamics, 41, 2973–3005.

. Russo, E. and Cubasch, U., 2016. *Mid-to-late Holocene temperature evolution and atmospheric dynamics over Europe in regional model simulations*, Climate of the Past, 12, 1645–1662.

**With kind regards on behalf of the all authors,**

**Emmanuele Russo**

Please also note the supplement to this comment:
https://www.geosci-model-dev-discuss.net/gmd-2019-22/gmd-2019-22-AC2-supplement.pdf
* * *
**[GMDD](https://www.geosci-model-dev-discuss.net/)**

Interactive
comment

[Figure]

**BIAS ($CCLM_{ERAInt}$ - $OBS$)**

[Figure]

**Fig. 1.** Mean bias of annual mean (left), winter mean (middle) and sum- mer mean (right) near surface temperature (T2M, ° C, top panel), precipi- tation (PRE, %, middle panel) and diurnal temperature range (D

**BIAS T2M DJF**

**REF-SNOW**

**REF-SOIL**

**Fig. 2.** Mean bias of near surface temperature (T2M, \textdegree C) winter values of the reference simulation \textbf{a} and the simulation \textbf{SNOW} (\textit{left}) and \textbf{SOIL} (\textit{right}) ,

BIAS ($CCLM_{dtchange} - OBS$)

**Fig. 3.** Maps of bias calculated for the NCEP2-driven simulation with the reference configuration but different time step, against different observational datasets for the 3 considered variables. Every row pre

RANGE OBS

T2M · YEAR · DJF · JJA

°C

0 5 10 15

PRE

%

0 20 40 60 80 100 120

DTR

°C

0 2 4 6 8 10 12

**Fig. 4.** Maps of the spread calculated among different observational datasets for each consid-
ered variable, for the annual mean (left), winter (middle) mean and summer mean (right). From
top to bottom, the va

**BIAS ($CCLM_{REF}$ - $OBS$) T 2M**

YEAR          DJF          JJA

°C

**Fig. 5.** Mean bias of annual mean (\textit{left}), winter mean (\textit{middle}) and summer mean (\textit{right}) near surface temperature (T2M, \textdegree C), of the reference COSMO-CLM simulation (a) with r

**BIAS** ($CCLM_{REF}$ - $OBS$) **DTR**

YEAR · DJF · JJA

CRU · ERAInt · MERRA2

**Fig. 6.** Mean bias of annual mean (\textit{left}), winter mean (\textit{middle}) and summer mean (\textit{right}) Diurnal Temperature Range (DTR, \textdegree C), of the reference COSMO-CLM simulation (a) with

**T2M**

**PRE**

**DTR**

**Fig. 7.** Skill Score (SS) derived from the weighted MAE ($MAE\_w$, Eq. \ref{eqa}) calcu-
lated over the monthly climatological values of the seasonal cycle of different COSMO-CLM
simulations and observational da

**SS - Subdomains**

**T2M (CRU)**

**PRE (GPCC)**

**DTR (CRU)**

**Fig. 8.** Skill Score (SS) derived from the MAE calculated for each domain sub-region over the monthly climatological values of the seasonal cycle of different COSMO-CLM simulations and observational datasets.

**SS - Subdomains**

**Fig. 9.** Skill Score (SS) derived from the MAE calculated for each domain sub-region over the monthly climatological values of the seasonal cycle of different COSMO-CLM simulations and observational datasets.

SS (Weighted MAE) – Subdomains

**T 2M (CRU)**

**PRE (GPCC)**

**DTR (CRU)**

**Fig. 10.** Skill Score (SS) derived from the weighted MAE ($MAE_w$, Eq. \ref{eqa}) calculated for each domain sub-region over the monthly climatological values of the seasonal cycle of different COSMO-CLM simula

[Figure]

**Fig. 11.** Orography map of the Central Asia simulation domain on a regular grid with a spatial resolution of 0.25 km. Masked in gray are the ocean and the external area of the domain.

---

## Author Comment (AC3) · 18 Jul 2019

Reply to
**2nd Reviewer**
*Russo, E., Kirchner, I., Pfahl, S., Schaap, M., and Cubasch, U.: Sensitivity studies with the Regional Climate Model COSMO-CLM 5.0 over the CORDEX Central Asia Domain, Geosci. Model Dev. Discuss., https://doi.org/10.5194/gmd-2019-22.*

[Figure]

Dear reviewer,

Thank you very much for your effort in reviewing our paper.

Below we go point by point through your technical corrections, presented in italic, detailing how we dealt with your concerns reported in *Bold*.

Please, also find attached as supplement another version of this response, where the captions of the figures are entirely reported.

Thank you.

*General Comments*

- *I*n order to provide reliable future climate projections, the model should be able to capture the present climate feature realistically. For seeking the optimal setups for regional climate model COSMO-CLM over the CORDEX Central Asia domain, the authors have conducted series of sensitivity simulations for historical periods. With different observation/reanalysis dataset as references, they evaluated the general model performance in capturing the mean climate and variability of temperature, precipitation and daily temperature range and figured out the relative optimal model setups for CORDEX Central Asia domain.

  Though the study is rather regional specific, it is believed to be interesting for the regional climate modelling community. The manuscript is in general well organized. The methods used are reliable and language is good. However, the manuscript suffers from some major problems. The authors will need to address them before the manuscript can be considered for publication in Geoscientific Model Development.

*Specific Comments*

- *I*t is suggested to reduce to a relative brief introduction about vulnerability of CORDEX Central Asia to the effects of climate changes, say from Page 2 Line 19 to P3 L11. Furthermore, there is a general lack of reviewing studies about model performance evaluation, which are related to the experimental setups, assessment methods and discussion, c.f., Li et al. (2018) and Huang et al. (2015) and so on: Li, D., Yin, B., Feng, J., Dosio, A., Geyer, B., Qi, J., ... Xu, Z. (2018). Present Climate Evaluation and Added Value Analysis of Dynamically Downscaled Simulations of CORDEX—East Asia. Journal of Applied Meteorology and Climatology, 57(10), 2317-2341. Huang, B., Polanski, S., Cubasch, U. (2015). Assessment of precipitation climatology in an ensemble of CORDEX-East Asia regional climate simulations. Climate Research, 64(2), 141-158.

**We agree with the referee that the part of the introduction on the vulnerability of Central Asia domain to the effects of climate change should sensibly be reduced, being only secondary to the purposes of the manuscript and making the text hard to read. We will try to summarize this part in a more concise way in the new version of the manuscript. Additionally, the part of the introduction on the state of the art of model performance evaluation and model calibration will be extended, considering, among others, the referee suggested references relevant for the area.**

- *T*he authors conducted a series of experiments considering different configurations, which are supposed to be significant for skills of modelling. However, some specific setups, which have been proved to be important in regional climate modelling, have not been considered in the study, such as the technique of spectral nudging (von Storch et al. 2000) and topography. RCM simulation with spectral nudging can add value in reproducing snow water equivalents, coastal winds

and some meso-scale phenomena (von Storch et al. 2016), as well as annual mean temperature and precipitation (Tang et al. 2017). The reviewer suggest the authors add one experiment with spectral nudging. In addition, about two additional 25-year long simulations covering 1991-2015, why do not use a period backward, say 1981-2005, so that there are longer spinup time, and same comparison period as other experiments?

**We agree with the referee that spectral nudging is a powerful tool in order to add value to several aspects of RCM simulations, as indicated in Von Storch et al. 2016 and Tang et al. 2017. Nevertheless, we think that the use of spectral nudging does not fit well the scopes of our work. In fact, in the paper we want to evaluate general model performance and how it is possible to improve these by using a set of specific physical configurations. Also, we want to determine main model limitations and uncertainties and the possible reasons for them. For doing this, we think that it is of fundamental importance to let the model "free" to develop. We do not think that constraining the model by spectral nudging would be useful in this sense. On top of that, this step is not considered in the main CORDEX-CORE directives and also in the model configuration procedure of the COSMO-CLM community. Concerning the point on why we performed the 25-year long simulations over the period 1991-2015, the response is that we aimed to use for this, the restart file of the reference simulation (01 January 2006). This allowed to save computational time, because otherwise the reference simulation should have been repeated for 25 years, starting at 1981 instead of 1991.**

- *T*here are some problems in Figure plottings: a). Figure 1, please plot in lon and lat dimensions rather than in rlon, rlat dimensions; b). Figure 2, it is better to add names of subregions on map rather than using a colorbar; c). Figure 3, the colorbar scheme is rather poor. It is hard to distinguish them on the map. Less and distinguishable colors are suggested to use, with more equal divisions within

-10 to 10 and less divisions from (plus minus) 10 to (plus minus) 20.

**We modified Fig.1 of the former version of the manuscript as suggested by the referee. We propose now to replace the former figure with Fig.?? of the current document. We also modified Fig. 2 of the former version of the manuscript accordingly to the referee comment. The new figure is shown as Fig. ?? of the current manuscript. We agree with the referee that this new figure might sensibly help improving the results discussion for different sub-regions. Finally, we also modified Fig.3 and 5 of the former version of the manuscript following the referee suggestion, reducing the number of colorbar breaks. We want to highlight the fact that the new figures, reported here as Fig.?? and Fig. ??, allow now to better discriminate high biases and in particular to notice that, for the case of winter temperatures, these are mainly inherent to the UDEL dataset and that, in general for temperature, biases exceeding 10 °C are only present for a few number of points for areas characterized by particularly complex topography.**

- *S*ome descriptions does not reflect the figures or tables. Such as P10 L26, I would not say experiment q in Fig.7 (upper panel) fits to the description; P10 L34, experiment o does not share the use of the setup of j. A thorough revision is needed to catch all these inconsistencies.

**Concerning the comment for page 10, line 26, we realize that we were not probably very clear in the description of the figure of seasonal calculated SS. Here we wanted to say that for temperature experiment q has positive values for all seasons, except winter. We will modify the text accordingly. Instead we agree with the referee comment relative to page 10, line 34, and we will try to revise the entire text for similar inconsistencies.**

- *I* would not agree the conclusion that "The results for the mean climate appear to be independent of the observational dataset used for evaluation and of the

boundary data employed to force the simulations". In fact, according to Fig. 3 and Fig. 5, it is clear that skill of simulated mean climate depends on the referred observational dataset. Furthermore, Li et al. JAMC (2018) clearly shows that both observational dataset and boundary forcing have impacts on the skill assessment of simulated mean climate.

**Following the referee comment we acknowledge the fact that the highlighted sentence was probably not very clearly expressed. What we wanted to say in this case was that when considering different observational datasets and different boundaries, in our case study, we see that experiment q leads to an univocal positive improvement of the simulated results, for all variables, in all the cases. Considering this point, we will re-formulate the highlighted part of the text in a clearer way.**

- *O*nly whole-region or subregion averaged values for SS or variance ration (Fig. 6 – Fig. 8) are not enough. Spatial distribution patterns of these scores are significant for a thorough model quality assessment. I would not suggest to plot every spatial distribution of these scores for each reference dataset, but representative figures are necessary, if not in the manuscript but in the supplementary part.

**We agree with the referee. A similar concern was also raised by the other referee. We agree on the fact that analyses on sub-regions for the climatological means could be very important for the purposes of the paper. For this we now propose to substitute the figure on the SS of the different simulations calculated for single seasons, with Fig.?? of the following document, placing the former in the supplement part of the paper. The new figure shows the SS of the MAE calculated over all the points of each sub-domain characterized by similar climatic conditions. This might help to distinguish different biases in different cases, and to determine how and to which degree it is possible to reduce them, through**

modification in specific physical settings of the model. On the other hand, we think that the analyses of variance are already in their definitive form. In fact, for this we proceeded in the same way as in Gleckler et al. 2008 and also considering Wilks 2006. Basically, the assumption that we follow is that the model, due to its chaotic nature, is not supposed to catch climate variability point by point. For this reason it is better to use regional means when we want to evaluate model variability. We will try to modify the text in order to make this point clearer.

*Minor Comments*

- *P*6 L8-15: It's better to summarize the data information in a table.

**We agree and we will add a table with information for the different observational datasets. Still, we think it is important to also mention these datasets in the text, together with appropriated references.**

- *P*7 L6: Tab. 3 not Tab.4, the same for P9 L6 and P12 L14

**We agree and will modify the text accordingly.**

- *P*7 L7-8 Combine two paragraphs into one

**We agree. We will join the two paragraphs accordingly to the referee comment.**

- *P*7 L13: 'Mean Absolute Error' to 'Mean Absolute Error (MAE)'

**We will modify the text accordingly to the referee comment.**

- *P*11 L24-25: It may be only appropriate when you run CCLM driven by similar high quality reanalysis datasets.

**Again, here we wanted to show that the model presents the same improvements for experiment q when using NCEP2 and mainly employed ERAInterim reanalysis. We decided to use NCEP2 instead of ERAInterim, cause their resolution is**

[Figure]

closer to the one of the GCMs ($\sim$ 200 Km) that we aim to use for CORDEX simulations. Despite this more than reasonable choice, we also considered ERAInterim driven simulations in our paper, to show that in the two cases we get almost the same results. Please, find more details concerning this point in the answer to the first referee. We will modify the final version of the manuscript in order to make this point clearer.

- *P*12 L3-19: Please indicate which subpanel of Figure 8 you are describing in the text.

We agree that the current description of the analysis of the variance is a bit confusing and will try to improve it in the final version of the manuscript, better specifying in each case the considered figure sub-panel, as suggested by the referee.

- *P*12 L26-27: range of absolute differences instead of absolute differences?

We agree. We will modify the text accordingly.

Below we propose some additional bibliography that we will provide in the revised version of the manuscript, if not already present, accordingly to the referee comments.

**References**

Li, D., Yin, B., Feng, J., Dosio, A., Geyer, B., Qi, J., ... Xu, Z., 2018. *Present Climate Evaluation and Added Value Analysis of Dynamically Downscaled Simulations of CORDEX—East Asia*, Journal of Applied Meteorology and Climatology, 57(10), 2317-2341.
, Huang, B., Polanski, S., Cubasch, U., 2015. *Assessment of precipitation climatology in an ensemble of CORDEX-East Asia regional climate simulations.*, Climate Research, 64(2), 141-158.

**With kind regards on behalf of the all authors,**

**Emmanuele Russo**

---

## Author Comment (AC4) · 18 Jul 2019

Dear reviewer,

we are sorry but the figures that we uploaded together with our response to your comments are not visualized.

For this reason we attached to our interactive response a supplementary version, including all figures.

With kind regards on behalf of the all authors, Emmanuele Russo

---

## Referee Report (RR1)

I found that the authors have addressed the comments I made in my previous review. I acknowledge the big effort made, especially in order to clarify the origin of the large biases, with even additional simulations.

However I cannot recommend yet the publication, being still some issues that need to be addressed. The most serious shortcoming is that the manuscript is messy and mixed up. There are many trivial errors revealing that the manuscript has not double checked carefully. The authors should check the manuscript before submission, in order not to worsen the work of the reviewer. More specifically, the figure numbering seems to be wrong when they are referenced. At page 7, line 32 you mention Fig.2, but Fig.2 shows the maps of the spread and not the subdomains. And so on… for almost all the cross-references. At page 11, line 13 you mention Fig.8 about SS analysis for subregions, but this figure seems missing so I cannot analyze the results described in Sec. 3.1.2.

**Other comments**

Pag 5, Lines 12: latidues ?

Pag 7, Lines 2-3: The authors decided not to address my previous comment about CDO, but I insist that this information is not relevant… otherwise you should mention also the software you have used for plotting the figures, for making calculations… However I leave the Editor the final decision about this comment.

Pag 10, line 6: observarions ?

Pag.11, lines 7-9 : "Additionally, the ranges… for the area". You claim that the range of improvements are similar, but sincerely the values reported in Table 4 seem to me quite different, keeping in mind that in a sensitivity context even a difference of +1 is a large value. More specifically 7.5 is different from 4.9, 11.1 is rather different from 7.4.

Sec. 3.1.4: probably in this case the cross reference of Fig. 8 is correct, but the figure and the text are not clear. What do these maps represent? You claim that SNOW configuration leads to warmer conditions, but I see a wide blue area on the right… probably I do not understand well the meaning of the figures.

Pag 12, line 30: WSH? Probably it is WSC

Pag 12, line 31: "The largest underestimation… CSA". I guess that this analysis is related to Fig.10, but I do not understand the meaning of this figure. Probably the quality of the colors is low.

Pag. 13, line 28: conlcusion ?

Pag 14, line 5: intiative ?

Pag 14, lines 20-23: "Following these… snow model". You say that no significant changes are evident. If so, what is the relationship with what you have written before (lines 11-20)?

Pag 14, line 24: I am not sure, but I guess that GUO can be written as Guo.

Figure 1 (caption). You claim that these are rotated coordinates, but they seem to me geographical coordinates.

Table 2: Why is there a question mark next to Tegen  ?

---

## Author Response (AR2)

Reply to
**Anonymous Referee**
*Russo, E., Kirchner, I., Pfahl, S., Schaap, M., and Cubasch, U.:*
*Sensitivity studies with the Regional Climate Model COSMO-CLM 5.0*
*over the CORDEX Central Asia Domain, Geosci. Model Dev.*
*Discuss., https://doi.org/10.5194/gmd-2019-22.*

Dear reviewer,

Thank you again for your effort in reviewing our paper.

Below we go point by point through your technical corrections, presented in *italic*, detailing how we dealt with your concerns reported in **bold**.

Thank you.

*General Comments*

- *I found that the authors have addressed the comments I made in my previous review. I acknowledge the big effort made, especially in order to clarify the origin of the large biases, with even additional simulations. However I cannot recommend yet the publication, being still some issues that need to be addressed. The most serious shortcoming is that the manuscript is messy and mixed up. There are many trivial errors revealing that the manuscript has not double checked carefully. The authors should check the manuscript before submission, in order not to worsen the work of the reviewer. More specifically, the figure numbering seems to be wrong when they are referenced. At page 7, line 32 you mention Fig.2, but Fig.2 shows the maps of the spread and not the subdomains. And so on... for almost all the cross-references. At page 11, line 13 you mention Fig.8 about SS analysis for subregions, but this figure seems missing so I cannot analyze the results described in Sec. 3.1.2.*

**Following the referee comment, we realized that we mixed up the numbering of the figures in the text of the previous version of the manuscript. We will correct all the figures numbering in the new version of the manuscript. Concerning Fig. 8 of the previous manuscript version, we were actually referring to fig. 9, that will now correctly referenced in section 3.1.2. Additionally, we reviewed the text more thoroughly, as suggested by the referee, correcting all the possible errors that we encountered.**

*Other Comments*

- *Pag 5, Lines 12: latidues ?*

  **Corrected.**

- *Pag 7, Lines 2-3: The authors decided not to address my previous comment about CDO, but I insist that this information is not relevant... otherwise you should mention also the software you have used for plotting the figures, for making calculations... However I leave the Editor the final decision about this comment.*

  **We finally agree with the referee and decided to exclude the citation to CDO from the final version of the manuscript.**

- *Pag 10, line 6: observarions ?*

  **Corrected.**

- *Pag.11, lines 7-9 : Additionally, the ranges... for the area. You claim that the range of improvements are similar, but sincerely the values reported in Table 4 seem to me quite different, keeping in mind that in a sensitivity context even a difference of +1 is a large value. More specifically 7.5 is different from 4.9, 11.1 is rather different from 7.4.*

  **We agree with the referee on the fact that some differences in the magnitude of the changes in SS are present. Nevertheless, this does not exclude the fact that the behavior of the model in the two cases is quite similar. This confirms the fact that in the two different cases the model will produce an important improvement. Following the referee comment, we tried now to be more careful in the final version of the manuscript, saying that the results are similar in the two cases, even though some changes are present.**

- *Sec. 3.1.4: probably in this case the cross reference of Fig. 8 is correct, but the figure and the text are not clear. What do these maps represent? You claim that SNOW configuration leads to warmer conditions, but I see a wide blue area on the right... probably I do not understand well the meaning of the figures.*

  **The previous figure 8 of the former manuscript version was actually misleading. This figure represented the bias of winter temperature between the reference simulations *a* and the SOIL and SNOW simulations. We now propose a new figure**

where the bias is calculated as **SOIL (SNOW)-REF** and where it is possible to see now that the **SNOW** simulation produces even warmer winter temperatures over Western Siberia. We have modified the figure caption accordingly. Additionally, realizing that the description of the figure in sec. 3.1.4, we tried to improve it in the new version of the manuscript.

- *Pag 12, line 30: WSH? Probably it is WSC*

  **Corrected.**

- *Pag 12, line 31: The largest underestimation... CSA. I guess that this analysis is related to Fig.10, but I do not understand the meaning of this figure. Probably the quality of the colors is low.*

  Fig. 10 of the former manuscript version shows the variance ratio calculated between the different experiments and the observations. We actually think that the figure was already pretty well described in the text and in the figure caption. Concerning the colorbar we think that the choice we have made was an optimal compromise to show the different changes in variance ratio among the different variables and regions characterized by completely different climate conditions. We think that the used colorbar allows to perfectly see the fact that different experiments do not really perform significantly different in terms of simulated variance; it also allows to easily distinguish for which variables and regions the model has good or bad performances in the simulation of variability. For the given reasons, also considering the fact that we did not receive any other comment about the figure from the 2nd reviewer, we think that the selected colorbar should not be changed. Nonetheless, acknowledging the referee comment, we revised the text and the figure caption to make its description a bit clearer.

- *Pag. 13, line 28: conlcusion?*

  **Corrected.**

- *Pag 14, line 5: intiative ?*

  **Corrected.**

- *Pag 14, lines 20-23: Following these... snow model. You say that no significant changes are evident. If so, what is the relationship with what you have written before (lines 11-20)?*

  **Following the reviewer comment, we reformulated the previous period emphasizing the fact that the results of these two additional simulations do not allow to improve model results. Considering also the model pretty low sensitivity in winter to the different changes in configuration, we can affirm that the model has some structural error that does not allow to properly simulate winter temperatures over Western Siberia and that new parameterizations for the simulation of processes related to snow cover and permafrost are required in COSMO-CLM.**

- *Pag 14, line 24: I am not sure, but I guess that GUO can be written as Guo.*

  **Corrected.**

- *Figure 1 (caption). You claim that these are rotated coordinates, but they seem to me geographical coordinates.*

  **Corrected.**

- *Table 2: Why is there a question mark next to Tegen?*

  **Following the referee comment we realized that we forgot to reference the TEGEN aerosol data set in Table 2 of the previous manuscript version. We will include the proper reference in the new manuscript.**

[revised manuscript text omitted]

---

## Author Response (AR3)

**Reply to Editor Comments on: Sensitivity studies with the Regional Climate Model COSMO-CLM 5.0 over the CORDEX Central Asia Domain**

*Emmanuele Russo*

Dear Editor,

thank you very much for your constructive comments to our manuscript entitled "Sensitivity studies with the Regional Climate Model COSMO-CLM 5.0 over the CORDEX Central Asia Domain". Below we go point by point through your technical corrections, reported in **bold**, detailing how we dealt with them in *italic*.

Thank you for the time and efforts you have put into your comments.

- **There are a few small issues that still need to correct before the paper can be accepted. What comes to Code and data availability, following the editorial guidelines (https://www.geosci-model-dev.net/12/2215/2019/gmd-12-2215-2019.pdf) you should make the used data available also either in supplementary material, separate doi or added to zenodo where the plot code is published. So simply having text that data available upon request is not applicable.**

  *Thank you very much for emphasizing the importance of transparency and reproducibility of scientific results. We now made all the scripts used in the postprocessing of the data and analyses presented in the paper freely available on ZENODO. At the same time, we made available on the same platform all the model configuration files used to run the presented experiments. These include both the files for the model simulations setup, and the ones employed in the COSMO-CLM interpolation routine INT2LM. Finally, we also shared all the model data used for the anlyses presented in the paper. The links to all these files have been added to the Code and data availability section of the manuscript.*

- **P4, L20: Comma missing from the reference before et al.** *We now modified the citation.*

- **P5, L5: Extra parenthesis around reference year** *We realized that we already referenced the COSMO-CLM in the introduction (P4, L21) and*

*consequently decided to remove the previous reference on P5, L5. Additionally, since COSMO-CLM is mentioned even earlier in the introduction, we moved its reference at P3, L25 of the new manuscript version.*

- **Remove "from their Web site at https://www.esrl.noaa.gov/psd/"** *We now removed the Web site link as suggested by the editor.*

- **P9, L17, L25 L27: space between Fig. 4 and Fig. 5 (and on next pages also)** *Following the editor comment, we corrected all similar errors throughout the text.*

- **P9, L29-20: extra spaces in the beginning and end of brackets.** *Corrected.*

- **Figure 1: Font quality not the best. Source for the figure not specified.** *We modified the figure and added in the corresponding caption the reference for the GLOBE dataset from where topography information is derived.*

- **Figure 2, 4, 5: Quality not good particularly what comes to fonts.** *Following the editor's comment, we realized that the quality of the previous figures was not optimal. In particular their fonts were highly blurred. We now tried to revise all the paper figures, trying to improve their quality and making all the labels and fonts more readable.*

- **Table 3: No lines in the table.** *We removed the table horizontal lines, as suggested by the editor. Nevertheless, for us is of fundamental importance to keep the different sets of experiments separated, as they are discussed in the paper. For this we inserted an empty line between the different sets of simulations in the new version of the table.*

---

## Author Response (AR4)

**Reply to Editor Comments on: Sensitivity studies with the Regional Climate Model COSMO-CLM 5.0 over the CORDEX Central Asia Domain**

*Emmanuele Russo*

Dear Editor,

we apologize for reporting the wrong link. We uploaded now the manuscript with the correct link to the data on ZENODO.

Best Regards,

Emmanuele Russo